# Strength of immune selection in tumors varies with sex and age

Andrea Castro [1,2,3,13], Rachel Marty Pyke[1,2,13], Xinlian Zhang[4], Wesley Kurt Thompson[4], Chi-Ping Day [5], Ludmil B. Alexandrov [6,7,8], Maurizio Zanetti[8,9,10] & Hannah Carter [1,2,8,11,12✉]

Individual MHC genotype constrains the mutational landscape during tumorigenesis. Immune checkpoint inhibition reactivates immunity against tumors that escaped immune surveillance in approximately 30% of cases. Recent studies demonstrated poorer response rates in female and younger patients. Although immune responses differ with sex and age, the role of MHC-based immune selection in this context is unknown. We find that tumors in younger and female individuals accumulate more poorly presented driver mutations than those in older and male patients, despite no differences in MHC genotype. Younger patients show the strongest effects of MHC-based driver mutation selection, with younger females showing compounded effects and nearly twice as much MHC-II based selection. This study presents evidence that strength of immune selection during tumor development varies with sex and age, and may influence the availability of mutant peptides capable of driving effective response to immune checkpoint inhibitor therapy.

[1] Department of Medicine, Division of Medical Genetics, University of California San Diego, La Jolla, CA 92093, USA. [2] Bioinformatics and Systems Biology Program, University of California San Diego, La Jolla, CA 92093, USA. [3] Health Science, Department of Biomedical Informatics, School of Medicine, University of California San Diego, La Jolla, CA 92093, USA. [4] Department of Family Medicine and Public Health, Division of Biostatistics & Bioinformatics, University of California San Diego, La Jolla, CA 92093, USA. [5] Laboratory of Cancer Biology and Genetics, National Cancer Institute, National Institutes of Health, Bethesda, MD 20892, USA. [6] Department of Cellular and Molecular Medicine, University of California San Diego, La Jolla, CA 92093, USA. [7] Department of Bioengineering, University of California San Diego, La Jolla, CA 92093, USA. [8] Moores Cancer Center, University of California San Diego, La Jolla, CA 92093, USA. [9] The Laboratory of Immunology, University of California San Diego, La Jolla, CA 92093, USA. [10] Department of Medicine, Division of Hematology-Oncology, University of California San Diego, La Jolla, CA 92093, USA. [11] Cancer Cell Map Initiative (CCMI), University of California San Diego, La Jolla, CA 92093, USA. [12] CIFAR, MaRS Centre, West Tower, 661 University Ave., Suite 505, Toronto, ON, Canada. [13] These authors contributed equally: Andrea Castro, Rachel Marty Pyke. ✉email: hkcarter@health.ucsd.edu

The major histocompatibility complex (MHC) exposes protein content on the cell surface to allow detection of antigens by the immune system. This applies to nonself antigens such as viral proteins, and self-proteins that include tumor antigens. Tumor cells harbor oncogenic alterations that can be presented to the immune system by the MHC, causing immune recognition and elimination (immune surveillance)[1]. However, in order to grow, invade, and spread, tumors must evade immune surveillance. Common mechanisms of immune evasion include loss of the MHC molecules and the upregulation of immune checkpoint molecules on cell surfaces that normally regulate the amplitude and duration of a T-cell response[2]. Immune checkpoint blockade (ICB) uses antibodies to block these immune checkpoint molecules, and can invigorate inactive and/or exhausted T cells to produce antitumor effects that confer long-term survival benefits in certain types of cancer[3]. However, ICB is effective in only 10–40% of patients for reasons that remain unclear. Meta-analyses of clinical trials in multiple cancer types treated with ICB suggest that young and female patients are characterized by low response rates[4–8]. The reason(s) for the poor response of these two populations remains elusive.

An accumulating body of literature points to sexual dimorphism in immune responses[9]. Moderated by genetic and hormonal factors, females have twice the antibody response to influenza vaccines[10] and higher CD4+ T-cell counts than males[11]. Moreover, females are far more susceptible to autoimmune diseases[12], demonstrating a stark imbalance in the way the immune response causes diseases in the two sexes. Immunosequencing of over 800 individuals revealed sex associated differences in the extent to which HLA molecules propagate selection and expansion of CD8+ T cells[13]. Interestingly, a stronger immune response in females has been observed across several species[14–16], and sexual dimorphism has been demonstrated in immune selection and restriction of intratumor genetic heterogeneity in a mouse model of B-cell lymphoma[17]. In addition, a recent study has found sex-based differences in molecular biomarkers and immune checkpoint expression in multiple tumor types treated with ICB[8]. Altogether, these studies suggest that these differences are sex-specific and not lifestyle dependent.

Studies have demonstrated age-related changes in immune response as well. As humans age, there is a decrease of general immune function including production of IL-2, a pivotal growth factor for T cells[18]. Reduced thymic output, lower numbers of naive T cells, and overall reshaping of the size and specificity of the T-cell repertoire by microbial pathogens may explain why, for example, about 90% of excess deaths during flu season occur in patients greater than 65 years of age[19]. In addition, elderly people have reduced phagocytic function and HLA-II expression on antigen presenting cells[20]. Collectively, these factors render elderly individuals less able to mount a T-cell response to new antigens and respond to vaccination.

Recently, we developed the Patient Harmonic-mean Best Rank (PHBR) score that quantifies patients' ability to present somatic mutations in their tumor by their specific MHC-I and MHC-II haplotypes[21,22]. PHBR-I and PHBR-II scores aggregate predicted peptide-MHC molecule binding affinities from established tools[23,24] to produce a mass spectrometry-validated, residue-centric, and patient-specific presentation score that captures a mutant peptide's visibility to the immune system. In previous publications we used PHBR scores to assess the role of MHC genotype in shaping mutation accumulation during tumorigenesis[21,22]. We found that patients tend to accumulate driver mutations that cannot be effectively presented by their own MHC molecules, likely a consequence of immune-based elimination of tumor cells harboring well-presented driver mutations, a selective process referred to as immunoediting[25]. This analysis revealed that thyroid carcinoma and low-grade glioma patients experience the highest MHC-based

selective pressure on driver mutations[21,22]. Interestingly, these tumor types also had the youngest average age at diagnosis compared to all studied tumor types. In light of these observations, we reasoned that younger and female patients may experience stronger immunoediting early in their tumor history, accumulating mutations that are less favorably presented by their MHC, i.e., mutations more invisible to their immune system, at the time of diagnosis. Predictably, a depletion of potentially immunogenic mutant peptides would cause ICB to be ineffective. At first approximation we ruled out an effect due to sex-specific (MHC-I Pearson R = 0.99, MHC-II Pearson R = 0.99) or age-specific (MHC-I Pearson R = 0.98, MHC-II Pearson R = 0.99) imbalances in MHC genotype frequencies. Therefore, we sought to test the hypothesis that sex- and age-specific differences in driver mutation presentation are the result of differential immunoediting.

In this study we find that female and younger patients exhibit stronger immune selection in their tumors, measured by the affinity of their observed, expressed driver mutations compared to male and older patients. MHC-II appears to have a stronger effect compared to MHC-I. Our findings, based on TCGA samples, are validated in an independent validation cohort.

## Results

**Fewer presentable drivers in female and younger patients.** We focused on a set of 1018 driver mutations, defined in[21], as driver mutations are more prevalent in the clonal architecture of an individual's cancer and confer a selective growth advantage. We assigned MHC-I and MHC-II types using PolySolver and HLA-HD, two exome-based calling methods[26,27] and considered only microsatellite-stable TCGA tumors. After excluding 515 patients from class I and 1064 patients from class II analyses due to HLA genotype incompatibility with NetMHCpan affinity prediction software, 9913 patients with MHC-I calls and 7174 patients with MHC-II calls remained. These patients were diverse in sex, with more males than females (Supplementary Fig. 1A), and a broad distribution of age at diagnosis (Supplementary Fig. 1B). PHBR-I and -II scores were calculated for all patients across the 1018 driver events by taking the harmonic mean of each allele's best NetMHCpan percentile rank affinity score, providing an estimate of each patient's potential to present each mutation via MHC-I and MHC-II, respectively. Importantly, the PHBR-I and PHBR-II scores aggregate percentile rank scores of mutated peptides relative to large numbers of random peptide provided by NetMHCpan-4.0 and NetMHCIIpan3.2. For single peptide-HLA pairs, percentile rank scores of 0.5% and 2% for MHC-I and 2% and 10% for MHC-II have been used to represent strong and weak binding cutoffs respectively[28,29].

To rule out other covariates, we performed a series of control analyses. We categorized patients into subgroups according to sex (male versus female) and age (younger versus older based on pan-cancer 30th and 70th percentiles at age of diagnosis for categorical analyses). For sex-specific analyses, we further excluded seven sex-specific tumor types (breast, cervical, ovarian, uterine, prostate, and testicular cancer). First, we established that there were similar average numbers of driver mutations across sex and age patient groups (Supplementary Fig. 2). We previously found that TCGA patients with somatic MHC-I mutations had altered mutational landscapes, with a higher fraction of binding mutant peptides than patients without MHC-I mutations[30]. To ensure that somatic MHC-I mutations would not skew the driver mutation PHBR-I score distributions, we compared scores for patients with and without MHC-I mutations grouped by sex and age and found no significant differences (Supplementary Fig. 3). We then compared the distributions of patient PHBR-I and PHBR-II scores across the 1018 driver mutations (Supplementary Fig. 4A–D) and found

significant *p* values, but very small effect sizes between groups. To ensure that the potential to present driver mutations was consistent across sex and age, we compared the fraction of presented drivers at various score thresholds, and found no significant differences (Supplementary Fig. 4E, F). The overall similarity of MHC presentation suggests that patients of both sexes and various ages at diagnosis present driver mutations with roughly equivalent efficacy, implying that specificity of MHC presentation resulting from specific allele combinations is not a mechanism causing differences in ICB response rate.

We therefore reasoned that the discrepancy might be due to differences in the strength of immune selection, e.g., tumors with stronger immunoediting should retain fewer driver mutations that are presentable to T cells by the patient's own MHC molecules. For sex- and age-specific groups in each cohort, we compared the PHBR-I and PHBR-II score distributions for observed, RNA-expressed driver mutations observed in patient tumors, excluding 4782 patients with no drivers from the list of 1018. While the number of observed drivers was not significantly different between sex and age groups (Supplementary Fig. 2), younger female patients were overrepresented in the group with no observed driver mutations (Fisher's exact test: class I: OR = 1.12, *p* < 0.12; class II: OR = 1.28, *p* < 0.015). We note this group had an overrepresentation of thyroid cancer cases, a disease associated with low mutational burden and that typically only has a single driver mutation[31]. We therefore performed sex-specific analysis for unique 2900 patients and age-specific analysis for 3928 unique patients.

Across pan-cancer cohorts, females were at a significant disadvantage (higher PHBR scores) in presenting their driver mutations by both their MHC-I and MHC-II molecules (Fig. 1a, b, *p* < 2.6e−04 and *p* < 1.2e−07, respectively). Younger patients also tended to have worse presentation of driver mutations by both MHC-I and MHC-II molecules (Fig. 1c, d, *p* < 2.4e−5 and *p* < 7.3e−04, respectively). Notably, the shift in PHBR score distributions between groups occurs near the threshold for weak binding. Given that a limited number of somatic mutations generate mutant peptides and not all of these are immunogenic, this small shift may translate to significantly less opportunity to generate a host antitumor response upon ICB. Importantly, we found that these observed between-group differences in PHBR scores were far greater (falling outside the 99% confidence interval) than differences when we randomly reassigned mutations across patients and recalculated patient-specific PHBR

scores (Methods; Supplementary Fig. 5), and were an order of magnitude greater than the effect sizes observed when comparing score distributions independent of mutation occurrence (Supplementary Fig. S4). We also found differences in affinity independent of the PHBR score, using median NetMHC-pan affinity scores across all alleles (Supplementary Fig. 6). Altogether this suggests that score differences do indeed result from the interaction of inherited MHC genotype with the observed mutations. Interestingly, the mutation-specific fraction of RNA reads mapping to these driver mutations was significantly lower for females and younger patients (Supplementary Fig. 7), further supporting sex- and age-based differential strength in immune selection.

We next examined evidence for sex and age differences in specific tumor types, adjusting age thresholds according to tumor type. There was a general trend for female and younger patients' tumors to have higher median PHBR-I and II scores across tumor types, although the difference was only statistically significant in melanoma (Supplementary Fig. 8A). We observed more variability in the trends across tumor types by age. Younger individuals trended toward higher median PHBR-I and II scores in tumors where the 30th/70th percentile was associated with a large age gap and the younger age threshold was under 55, with some notable exceptions that included rectal cancer, thyroid cancer, stomach cancer, and liver (Supplementary Fig. 8B). Overall these trends suggest that stronger pan-cancer immune selection in younger and female patients results broadly from effects observed across multiple tumor types.

Next, we explored the effect of age and sex in the context of the immune system's ability to eliminate effectively-presented mutations by modeling the relationship between mutation occurrence and immune visibility as modeled by PHBR-I and II scores. We constructed sex- and age-specific generalized additive models with random effects to account for variation in mutation rate across individuals, and examined the coefficients corresponding to independent and interaction effects for PHBR-I, PHBR-II, and sex or age to assess their contribution to immune selection for expressed mutations observed ≥2 times in the cohort, excluding patients with no observed, expressed driver mutations. To control for the fact that some driver mutations occurred in the same tumor, and thus are not completely independent events, we included patient ID as a random effect in our linear model. In both models, we found that PHBR-I and PHBR-II scores alone had significant effects on the probability of a mutation to be a

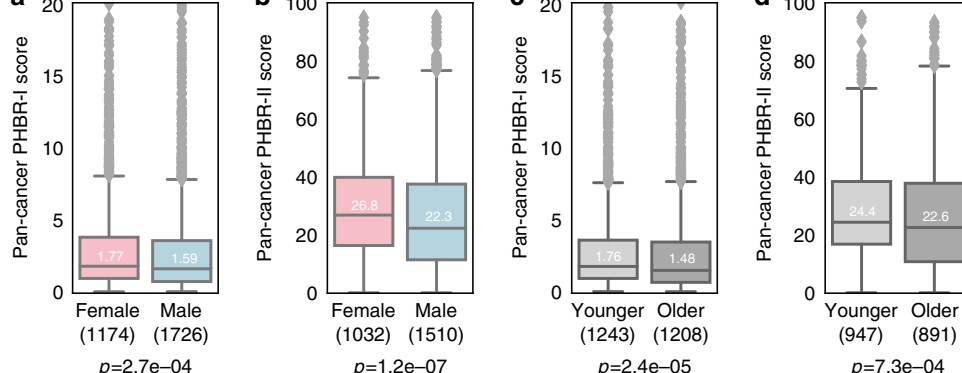

**Fig. 1 Sex- and age-specific MHC presentation of observed, RNA-expressed driver mutations. a**, **b** Box plots denoting the distribution of (**a**) PHBR-I and (**b**) PHBR-II scores for expressed driver mutations in female and male pan-cancer patients. **c**, **d** Box plots denoting the distribution of (**c**) PHBR-I and (**d**) PHBR-II scores for expressed driver mutations in younger and older pan-cancer patients. *P* values were calculated using the one-tailed Mann–Whitney *U* test. Median values are shown in each boxplot. All box plots include the median line, the box denotes the interquartile range (IQR), whiskers denote the rest of the data distribution and outliers are denoted by points greater than ±1.5 × IQR. The following effect sizes were calculated using Cliff's d: (**a**) *r* = −0.0654, (**b**) −0.104, (**c**) −0.081, (**d**) −0.0734.

**Table 1 Quantitative estimate of the association between PHBR score and mutation occurrence in sex- and age-specific cohorts.**

| | Parametric coefficients | Estimate | Pr(>\|z\|) |
|---|---|---|---|
| Sex analysis | **PHBR-I** | **0.048** | **0.0035** |
| | **PHBR-II** | **0.31** | **1.66e−56** |
| | Sex | −0.02 | 0.59 |
| | **PHBR-I:sex** | **0.07** | **0.02** |
| | **PHBR-II:sex** | **0.15** | **0.00035** |
| Age analysis | **PHBR-I** | **0.043** | **0.0078** |
| | **PHBR-II** | **0.31** | **1.01e−54** |
| | Age | −0.0025 | 0.06 |
| | **PHBR-I:age** | **−0.0029** | **0.005** |
| | **PHBR-II:age** | **−0.0035** | **0.007** |

Estimates and p values are shown for a generalized additive model with random effects relating PHBR scores to the set of expressed driver mutations observed ≥2 times in this cohort. P values were calculated via Wald tests using the Bayesian covariance matrix for the coefficients. Variables and their respective estimates and p values have been bolded if significant ($p < 0.05$).

target of immune selection (Table 1). Positive coefficients for both PHBR scores indicate that the higher the PHBR score (i.e., poorer presentation), the higher the probability of mutation. Furthermore, when we quantified the influence of both scores on probability of mutation using odds ratios between respective 25th and 75th percentiles, we found that PHBR-II (OR: 3.4, CI [3.19, 3.6]) has a much larger impact on probability of mutation than PHBR-I (OR: 1.27, CI [1.26, 1.29]), echoing the larger effect sizes seen in Fig. 1. As expected, sex and age alone did not influence the probability of mutation; however, of particular interest are the interaction terms that indicate the influence of PHBR scores on probability of mutation within the context of sex and age. Both the PHBR-I:sex and PHBR-I:age interactions as well as the PHBR-II:sex and PHBR-II:age interactions were significant. The negative PHBR:age estimates indicate stronger effects of PHBR-I as well as PHBR-II contribution to the probability of mutation in younger patients. On the other hand, positive PHBR:sex estimates indicate stronger effects of PHBR-I and PHBR-II contributing to probability of mutation in females according to the model formulation (Methods). Collectively, these results suggest stronger immune selection in females and younger patients.

As females and younger patients both demonstrated stronger immune selection compared to males and older patients, we further partitioned the cohorts simultaneously by sex and age, and investigated the distribution of PHBR-I and -II scores for these groups. We found that sex and age effects are cumulative, with tumors in younger females exhibiting significantly higher selective pressure by MHC than those in the other three groups (Fig. 2). We noticed a profound difference between PHBR score distributions between younger females and older males. Because younger males had worse presentation of their driver mutations compared to older females (Fig. 2), we sought to ensure that sex had an effect on immune selection independent of age. In two models incorporating sex, age, and PHBR-I and PHBR-II scores, respectively, both PHBR:sex and PHBR:age were independently significant for both class I and class II (Supplementary Table 1). These results demonstrate that more aggressive immune selection in younger females selects for tumors with driver mutations that are less visible to the immune system.

**Mutational signatures do not explain differential selection.** We next explored whether sex- and age-specific effects could be driven by differences in environmental exposure rather than

the strength of immune selection. Mutational signatures assign specific mutations to different mutagenic processes, allowing the exploration of differences in environmental exposure across sex and age. We compared the sex-specific occurrence of mutational signatures in each tumor type and found only a minority of instances where signature strength was weakly but significantly associated with sex (Fig. 3a). Importantly, only three of the signatures (01, 02, and 05) where we observed significant sex-specific differences contribute to the set of driver mutations used for this analysis (Fig. 3b). Since signatures 01 and 05 are endogenous rather than exposure associated signatures, this suggests a very low impact of environmental exposures on sex-specific effects of immune selection on drivers. Furthermore, when we excluded the tumor types with significant signature differences (glioblastoma multiforme, GBM and liver hepatocellular carcinoma, LIHC), we still observed sex- and age-related differences (Supplementary Table 2). In addition, only two signatures correlated with age, both of which have known association with aging[32]. We examined C>T and T>C mutations, which are hallmarks of signature 01 and 05, respectively, and found that observed driver mutations in these categories were broadly distributed across age at diagnosis. To explain weaker immune selection in older individuals, age-related mutations would have to be better presented (have lower PHBR scores) than other mutations. Instead, we found that C>T and T>C mutations were significantly more poorly presented (had slightly higher PHBR scores) than other mutations across all possible MHC-I and MHC-II alleles, suggesting that these mutations, and by extension, signatures 01 and 05, could not drive the apparent age-associated difference in immune selection (Fig. 3c). Thus, we conclude that the sex- and age-specific effects on immune selection are not likely due to environmental exposure differences[32,33].

**Validation in an independent non-TCGA cohort.** We sought validation of our findings in a cohort of 342 patients (309 with compatible MHC-I type calls and 277 with MHC-II type calls) compiled from published dbGaP studies and non-TCGA samples in the International Cancer Genome Consortium (ICGC) database[34] and filtered to exclude tumor types not represented in TCGA. While fewer tumor types were represented relative to the discovery cohort, these patients were diverse with respect to sex and age at diagnosis, with slightly more males than females, and similar average numbers of driver mutations. As in the discovery cohort, we found some significant differences in patient PHBR score distributions across the 1018 driver mutations, also with very small effect sizes between groups. Likewise, there was no difference in the fraction of presented drivers at various score thresholds (Supplementary Fig. 9). The majority of our validation cohort did not have expression data, so we predicted RNA expression using a logistic regression classifier trained on the TCGA cohort (Methods).

We found, as in the discovery cohort, that effectively-presented driver mutations were significantly depleted in younger and female patients compared to older and male patients (Fig. 4a–d). These differences were an order of magnitude greater than the effect sizes observed when comparing score distributions independent of mutation occurrence (Supplementary Fig. S9E–H).

When we examined the simultaneous effects of sex and age (Fig. 4e, f), younger females once again had significantly worse presentation of their driver mutations than older males across both MHC-I and MHC-II ($p < 0.001$, $p < 0.007$). We repeated the sex- and age-specific analyses using the generalized additive models and found that, for both sex and age, PHBR-II scores alone significantly influenced the probability of mutation, with

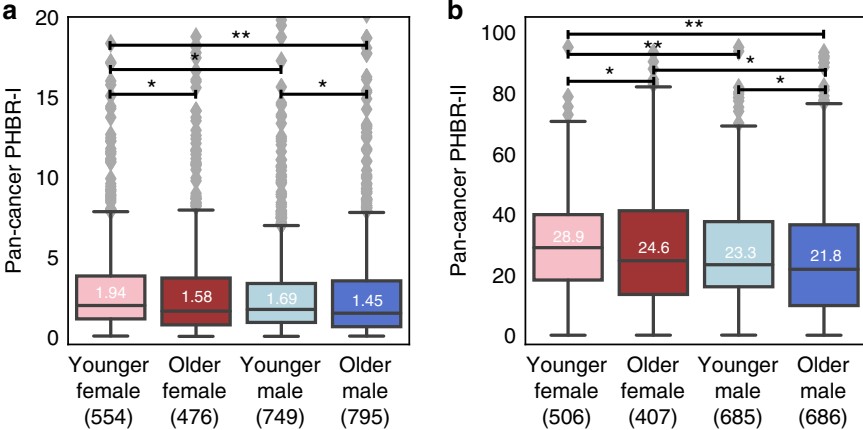

**Fig. 2 Integrated sex- and age-specific analysis. a** PHBR-I and **b** PHBR-II scores for the observed driver mutations in pan-cancer integrated sex- and age-specific patient cohorts. One asterisk indicates $p$ values < 0.05 and two asterisks indicates $p$ values < 0.001. All $p$ values were calculated using a one-tailed Mann–Whitney $U$ test. The Benjamini–Hochberg method was used to adjust for multiple comparisons for (**a**, **b**). Median values are shown in each boxplot. Exact $p$ values for (**a**) include: YF, OM: 0.7e−05; YF, OF: 0.005; YF, YM: 0.008; YM, OM: 0.008; OF, OM: 0.08; OF, YM: 0.22. Exact $p$ values for (**f**) include: YF, OM: 5.51e−07; YF, YM: 0.0003; YM, OM: 0.035; YF, OF: 0.038; OF, YM: 0.17. Y = younger, O = older, F = female, M = male. All box plots include the median line, the box denotes the interquartile range (IQR), whiskers denote the rest of the data distribution and outliers are denoted by points greater than ±1.5 × IQR.

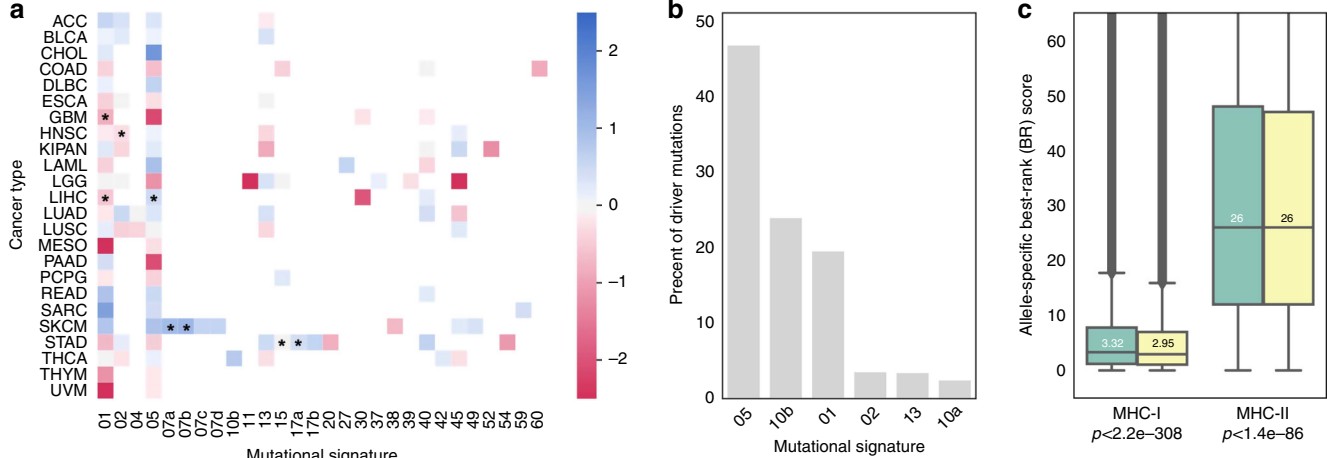

**Fig. 3 Sex-specific exposure analysis with mutational signatures. a** Heatmap of log2 male (blue) to female (pink) ratios of mutational signatures for each tumor type with asterisks denoting a significantly different ratio between male and female sexes. **b** The percentage of mutations in the set of driver mutations that are part of each mutational signature. **c** Boxplot comparing MHC-I and MHC-II presentation scores across all possible alleles for C>T or T>C driver mutations (green) versus driver mutations resulting from other base substitutions (yellow); 1,063,975 and 2,051,300 affinity scores were evaluated for C>T or T>C mutations for class I and II, respectively; and 1,851,025 and 3,568,700 affinity scores were evaluated for other mutations for class I and II, respectively. Exact $p$ values were calculated using a one-tailed Mann–Whitney $U$ test: (**c**) 2.2e−308 and (**d**) 1.4e−86. Median values are denoted in each boxplot. All box plots include the median line, the box denotes the interquartile range (IQR), whiskers denote the rest of the data distribution and outliers are denoted by points greater than ±1.5 × IQR.

higher PHBR scores (i.e., worse presentation) leading to higher probability of mutation (Supplementary Table 3). While PHBR-II:sex and PHBR-II:age coefficients trended in the same direction, with stronger effects in females and younger patients, they did not reach significance, likely due to sample size.

## Discussion

Here, we present evidence that both sex and age impact the driver mutations that arise and persist during tumorigenesis. We found that younger and female patients accumulate driver mutations in their tumors that are less readily presented by their MHC molecules (Fig. 5), suggesting a stronger toll by immune selection early in tumorigenesis. This finding is consistent with recent meta-analyses across multiple tumors showing sex- and

age-dependent differences in response to ICB[4–7]. We also observed the strongest effects in MHC-II based selection, in agreement with the fact that females have higher CD4+ T-cell counts than males[35]. A prevalent role of MHC-II driven immune selection can be explained by the fact that CD4+ T cells, besides direct effector function comparable to that of CD8+ T cells, also play a deep-rooted regulatory role in cooperating with CD8+ T cells via associative recognition of antigen[36,37]. Their function in orchestrating T-cell immunity, in general terms, makes them privileged actors, hence targets of immune selection as revealed herein. In older individuals, immune selection effects by MHC-II presentation of driver mutations are mitigated by a reduced CD4+/CD8+ ratio[38] and greater telomere attrition in CD4+ T cells than in CD8+ T cells[39] leading to accelerated senescence. Taken together, the

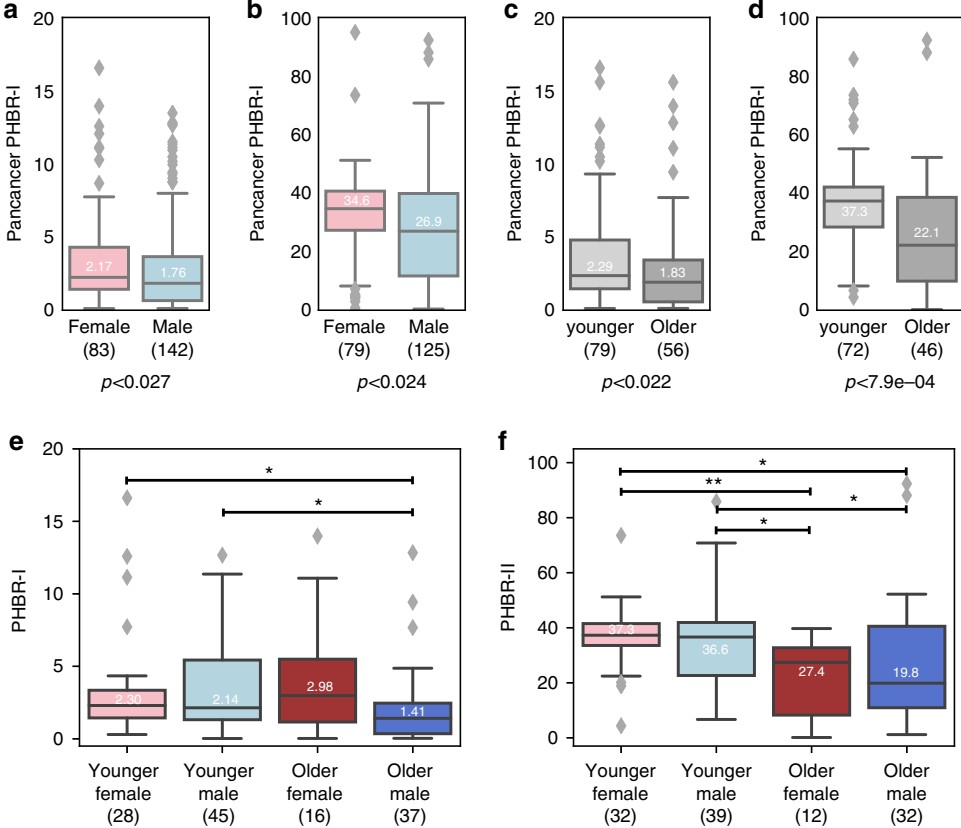

**Fig. 4 Sex- and age-specific MHC presentation of observed driver mutations in the validation cohort. a**, **b** Box plots denoting the distribution of (**a**) PHBR-I and (**b**) PHBR-II scores for driver mutations in female and male pan-cancer patients. Exact p values were calculated using a one-tailed Mann–Whitney U test: (**a**) 0.027 and (**b**) 0.024, and effect sizes were calculated using Cliff's d: (**a**) r = −0.154, (**b**) r = −0.164. **c**, **d** Box plots denoting the distribution of (**c**) PHBR-I and (**d**) PHBR-II scores for driver mutations in younger and older pan-cancer patients. Exact p values were calculated using a one-tailed Mann–Whitney U test: (**c**) 0.022 and (**d**) 7.9e−04, and effect sizes were calculated using Cliff's d: (**c**) r = −0.207, (**d**) −0.346. **e**, **f** Box plots denoting the distribution of (**e**) PHBR-I and (**f**) PHBR-II scores for driver mutations among integrated sex- and age-specific pan-cancer patient cohorts. One asterisk indicates p values < 0.05 and two asterisks indicates p values < 0.001. P values were calculated using a one-tailed Mann–Whitney U test. The Benjamini–Hochberg method was used to adjust for multiple comparisons for (**e**, **f**). Median values are shown in each boxplot. Exact p values for (**e**) include: YM, OM: 0.024; YF, OM: 0.028; OF, OM: 0.070; YF, OF: 0.56; YF, YM: 0.49; OF, YM: 0.50. Exact p values for (**f**) include: YF, OF: 0.0083; YF, OM: 0.013; OF, YM: 0.023; YM, OM: 0.045; YF, YM: 0.24; OF, OM: 0.34. Y = younger, O = older, F = female, M = male. All box plots include the median line, the box denotes the interquartile range (IQR), whiskers denote the rest of the data distribution and outliers are denoted by points greater than ±1.5 × IQR.

evidence suggests that tumors developing in younger and female patients are prone to stronger immunoediting than those in older and male patients.

Our findings based on the TCGA were reproduced in the smaller validation cohort where we once again observed poorer MHC-based presentation of driver mutations in females versus males and younger versus older patients, with presentation being worse in younger and female patients. When modeling the influence of MHC genotype on the probability of observing driver mutations, the estimated effect sizes are modest, although relatively large compared to effects detected by genome wide association studies where odds ratios are often <1.2[40]. Several sources of uncertainty, including errors in patient genotyping, prediction of the peptide-HLA binding affinities used to calculate the PHBR score, and errors in somatic mutation calling could obscure the true effects[21]. More accurate estimates will likely require larger sample sizes, and ideally availability of expression data as non-expressed mutations should not reflect the effects of immune selection.

In this analysis, we focused on a set of recurrent missense and indel mutations in established driver genes developed in our previous work. This is motivated by the assumption that these are

more likely to occur early during tumorigenesis, and may thus provide a view of immune selection before various mechanisms of immune evasion occur[22]. However it is unlikely that immune selection operates differently on different categories of mutation, and nondriver mutation-derived neoantigens should be equally capable of triggering a T-cell response. Whether tumor cells can evade T-cell responses more easily when they are targeted against nonessential nondriver mutations remains an important question. It has been suggested that ICB responses are most effective when a clonal driver neoantigen is present[41]. While we did not observe large sex or age bias in the mutational signatures associated with the 1018 driver mutations, we speculate that it is possible non-driver mutations could show differences in their potential to serve as neoantigens if the underlying mutational processes are active at different times or are biased to generate mutations in expressed protein coding sequences with characteristics that bias their presentation.

Notwithstanding some limitations, our analysis provides a compelling case for the paradigm that immune selection exerts its toll differently with respect to sex and age, with a greater effect in younger females. Of note, the younger female cohort had the poorest driver mutation presentation across both the discovery

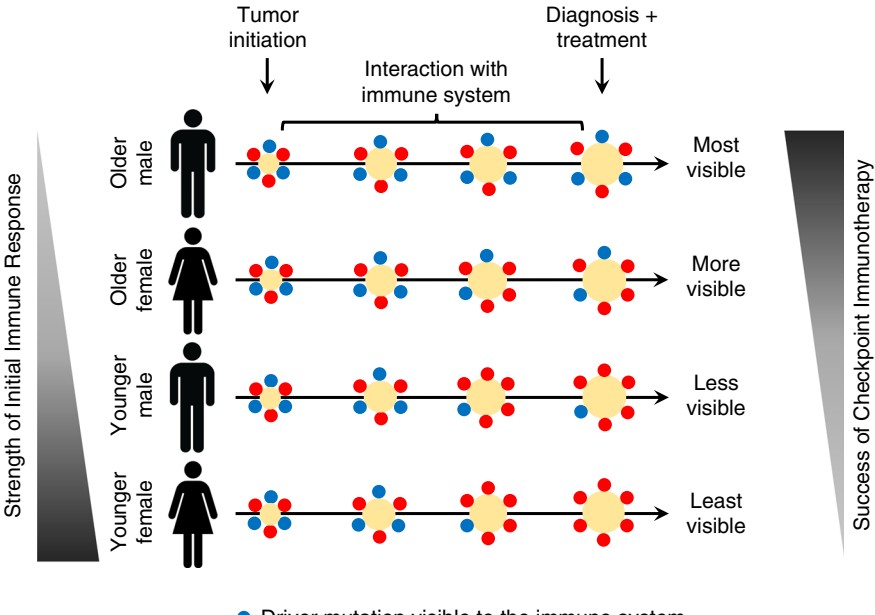

**Fig. 5 Proposed model of the relationship between immune selection and immunotherapy in cancer patients.** Young females experience the strongest immune response, rendering their diagnosed tumors more invisible to the immune system and difficult to treat with ICB. On the other extreme, old males experience the weakest immune response, leaving their diagnosed tumors more visible to the immune system and open to attack when stimulated with ICB. Blue dots indicate immunologically visible driver mutations while red dots indicate immunologically invisible driver mutations at various time points.

and validation cohorts, suggesting that these effects are strong and complementary. Although our analysis suggests that younger age is associated with stronger antitumor immune responses, we strongly suggest caution in considering whether this trend could generalize to pediatric tumors. The genomic landscape of pediatric tumors is distinct from that of adulthood tumors, with lower mutation burdens, different driver events and more germline factors and the characteristics of the pediatric immune system differ greatly from those of an adult[42]. Furthermore, we are unable to control for other sex- and age-related factors beyond predicted MHC presentation of driver mutation-derived peptides. These possibilities may include (a) differences in the antigen processing machinery preceding surface exposure of MHC-peptide complexes, and (b) genetic and epigenetic factors causing preferential mutation accumulation in the cohorts for reasons other than immunoediting.

In conclusion, this study indicates that immune selection exerts its toll differently with respect to sex and age, with a greater effect in younger females. As such, the response rate to ICB may be dependent on the strength of immune selection occurring early in tumorigenesis. Methods to accurately predict the impact of immunoediting on a patient-specific basis may lead to better predictive algorithms for response to therapy. As a corollary, we posit that ICB treatment is likely to have a reduced effect in younger female patients since this treatment will attempt to reactivate T cells for immunologically invisible neoantigens. Rather, adaptive T-cell therapy against patient-validated neoantigens or therapeutic vaccination against conserved antigens will likely be more beneficial in these patients. Notably prior to treatment with ICB, male sex (and less consistently older age) are associated with higher risk of recurrence and death in melanoma and may stand to benefit more from ICB[43,44], thus it is also possible that overall stronger immune surveillance could prove advantageous in the context of ICB despite differences in the quality of neoantigens. Finally, these findings shed light on the role of immune surveillance in cancer progression.

## Methods

**HLA typing.** HLA genotyping was performed for class I genes *HLA-A, HLA-B, HLA-C*, and class II genes *HLA-DRB1, HLA-DPA1, HLA-DPB1, HLA-DQA1*, and *HLA-DQB1*, which encode three protein determinants of MHC-I peptide binding specificity, *HLA-DR, HLA-DP*, and *HLA-DQ*. TCGA samples were typed with Polysolver[26], with default parameters, for class I and typed with HLA-HD[27], using default parameters, for class II. Both tools require germline (whole blood or tissue matched) whole exome sequenced samples. Samples with very low coverage on specific genes are left untyped by HLA-HD. Patients were assigned an *HLA-DR* type if they were successfully typed for *HLA-DRB1*. Patients were assigned *HLA-DP* and -*DQ* types if they had successful typing for *HLA-DPA1/HLA-DPB1* and *HLA-DQA1/HLA-DQB1*, respectively. Class I and class II types were validated by xHLA[45], run with default parameters, and only patients where all alleles agreed in both classes were included in the analysis.

**Presentation score assignment.** We used patient presentation scores, as defined in[21], to represent a particular patient's ability to present a residue given their distinct set of HLA types. For class I, 6 HLA alleles were considered (*HLA-A, HLA-B*, and *HLA-C*). For class II, 12 HLA-encoded MHC-II molecules (4 combinations of *HLA-DPA1/DPB1* and *HLA-DQA1/DQB1*; 2 alleles of *HLA-DRB1* considered twice each—since *HLA-DRA1* is invariant—for consistency between resulting molecules). NetMHCpan4.0[28] and NetMHCIIpan3.2[29] were used to calculate binding affinities. The PHBR score was assigned as the harmonic mean of the best residue presentation scores for each group of MHC-I and MHC-II molecules. A lower patient presentation score indicates that the patient's MHC molecules are more likely to present a residue on the cell surface.

**Set of driver mutations.** Somatic mutations were considered to be recurrent and oncogenic if they occurred in one of the 100 most highly ranked oncogenes or tumor suppressors described by Davoli et al.[46] and were observed in at least three TCGA samples. Among these, we retained only mutations that would result in predictable protein sequence changes that could generate neoantigens, including missense mutations and inframe indels. A total of 1018 mutations (512 missense mutations from oncogenes, 488 missense mutations from tumor suppressors, 11 indels from oncogenes and 7 indels from tumor suppressors) were obtained[21].

**Modeling the effects of PHBR score on mutation probability.** We built two matrices, for PHBR-I scores and PHBR-II scores, from the 1018 mutations and the 1912 patients with both PHBR-I and -II calls. Next, we built a binary mutation matrix $y_{ij} \in \{0,1\}$ indicating whether patient $i$ has a specific mutation $j$. We evaluated the relationship between this binary matrix, the matched $1912 \times 1018$ matrices with log PHBR-I and -II scores, $x1_{ij}$ and $x2_{ij}$, respectively, and the variable

of interest (sex or age) for patient $i$ and mutation $j$. We fit a generalized additive model for the centered log PHBR-I, centered log PHBR-II scores, centered sex (coded 0/1 for males/females) or centered age, and mutation probability with the GAM function in the MGCV R package[47]. To estimate the effects of PHBR and sex or age on probability of mutation, we considered the following random effects models:

$$\text{Logit}\left(\text{P}\left(y_{ij} = 1\right)\right) = \beta_1 x1_{ij} + \beta_2 x2_{ij} + \beta_3 \text{Sex}_i + \beta_4\left(x1_{ij} \times \text{Sex}_i\right) \\ + \beta_5\left(x2_{ij} \times \text{Sex}_i\right) + \eta_i, \tag{1}$$

$$\text{Logit}\left(\text{P}\left(y_{ij} = 1\right)\right) = \beta_1 x1_{ij} + \beta_2 x2_{ij} + \beta_3 \text{Age}_i + \beta_4\left(x1_{ij} \times \text{Age}_i\right) \\ + \beta_5\left(x2_{ij} \times \text{Age}_i\right) + \eta_i. \tag{2}$$

And PHBR-I and PHBR-II specific models (results in Supplementary Table 1):

$$\text{Logit}\left(\text{P}\left(y_{ij} = 1\right)\right) = \beta_1 x1_{ij} + \beta_2 \text{Age}_i + \beta_3 \text{Sex}_i + \beta_4\left(x1_{ij} \times \text{Sex}_i\right) \\ + \beta_5\left(x1_{ij} \times \text{Age}_i\right) + \eta_i, \tag{3}$$

$$\text{Logit}\left(\text{P}\left(y_{ij} = 1\right)\right) = \beta_1 x2_{ij} + \beta_2 \text{Age}_i + \beta_3 \text{Sex}_i + \beta_4\left(x2_{ij} \times \text{Sex}_i\right) \\ + \beta_5\left(x2_{ij} \times \text{Age}_i\right) + \eta_i. \tag{4}$$

where $\eta_i \sim N(0, \theta_\eta)$ are random effects capturing different mutation propensities among patients, using patient IDs. In these models $\beta_n$ measures the effect of the log-PHBR-I, log-PHBR-II, and sex or age. This analysis was repeated for the validation cohort.

**Mutational signature analysis.** Mutational signatures analysis was performed using a previously developed computational framework SigProfiler[48]. A detailed description of the workflow of the framework can be found in ref. [48], while the code can be downloaded freely from: https://www.mathworks.com/matlabcentral/fileexchange/38724-sigprofiler.

**Predicting RNA expression from DNA variant allelic fraction.** To predict binary RNA expression (≥5 reads at the mutant allele), we used the LogisticRegressionCV function from the Python sklearn v0.20.3 package to train a logistic classifier on the TCGA discovery cohort, using DNA variant allelic fraction (VAF), VAF percentile rank within the patient, and mutated gene as features. We conducted 10-fold cross-validation, achieving a mean 72% area under the receiver operating curve.

**Statistical analysis.** All box plots were evaluated using the default one-tailed Mann–Whitney $U$ statistical test, via the scipy.stats Python package. Mutational signature sex-specific distributions were also compared using the one-tailed Mann–Whitney $U$ test, and $p$ values were adjusted using the Benjamini–Hochberg Procedure. All boxplot figures include the median line, the box denotes the interquartile range (IQR), whiskers denote the rest of the data distribution and outliers are denoted by points determined by $\pm 1.5 \times \text{IQR}$. Effect sizes were calculated using Cliff's d (Cliff 1993).

## Data availability
Discovery cohort: data were obtained from publicly available sources including The Cancer Genome Atlas (TCGA) Research Network [http://cancergenome.nih.gov/]. TCGA normal exome sequences and TCGA clinical data were downloaded from the GDC on June 23–26th, 2018 and April 25th, 2017, respectively, using the gdc-client v1.3.0. Furthermore, TCGA somatic mutations were accessed from the NCI Genomic Data Commons [https://portal.gdc.cancer.gov/] on May 14th, 2017. Validation cohort: dbGaP studies (accession numbers: phs001493.v1.p1.c2, phs001041.v1.p1.c1, phs001425.v1.p1.c1, phs001493.v1.p1.c1, phs000980.v1.p1.c1, phs001469.v1.p1.c1, phs000452.v2.p1.c1, phs001451.v1.p1.c1, phs001519.v1.p1.c1, phs001565.v1.p1.c1) were obtained from the dbGaP database using the ascp tool from AsperaConnect v3.9.5.172984 and WXS/WGS data obtained from the Sequence Read Archive (SRA)[49] using the SRA toolkit v2.9.2. Somatic mutation files were obtained from the respective papers associated with each study. Additional non-TCGA patients' WXS/WGS data was obtained from the ICGC using the EGA download client v2.2.2 and icgc-get v0.6.1 and somatic mutation data from the ICGC DCC Data Release [https://dcc.icgc.org/] on (April 2, 2019 (PCAWG), March 18, 2019 (THCA-SA)) (Supplementary Dataset 1). The validation cohort's MHC-I and -II genotypes were typed using HLA-HD[27] and PHBR scores calculated using the method described in "Presentation score assignment". All remaining relevant data are available in the article, Supplementary Information, or from the corresponding author upon reasonable request.

## Code availability
Code to reproduce findings and figures can be freely accessed at https://github.com/CarterLab/HLA-immunoediting.

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

## Acknowledgements

We would like to thank T. Cameron Waller, Tina Wang, and Trey Ideker for scientific discussion. This work was supported by an NIH National Library of Medicine Training Grant T15LM011271 to A.C. an NSF graduate fellowship #2015205295 to R.M.P., NIH grants DP5-OD017937, an Emerging Leader Award from The Mark Foundation for Cancer Research, grant #18-022-ELA and a CIFAR fellowship to H.C. and RO1 CA220009 to M.Z. and H.C., P41-GM103504 for computing resources provided by the National Resource for Network Biology (NRNB). We would like to thank the TCGA research network for providing data used in the analyses, the ICGC database, as well as the following studies used in the validation cohort. *phs001493.v1.p1.c2 and phs001451.v1.p1.c1* We would also like to thank the Blavatnik Family Foundation, grants from the Broad Institute SPARC program, the National Institutes of Health (NCI-5R01CA155010-02, NHLBI-5R01HL103532-03, NCI-SPORE-2P50CA101942-11A1, NCI-R50-RCA211482A), the Francis and Adele Kittredge Family Immuno-Oncology and Melanoma Research Fund, the Faircloth Family Research Fund, and the DFCI Center for Cancer Immunotherapy Research fellowship and Leukemia and Lymphoma Society. *phs001041.v1.p1.c1.* We thank Martin Miller at Memorial Sloan Kettering Cancer Center (MSKCC) for his assistance with the NetMHC server, Agnes Viale and Kety Huberman at the MSKCC Genomics Core, Annamalai Selvakumar and Alice Yeh at the MSKCC HLA typing laboratory for their technical assistance, and John Khoury for assistance in chart review. *phs001425.v1.p1.c1* Christine N. Spencer, Pei-Ling Chen, Michael T. Tetzlaff, Michael A. Davies, Jeffrey E. Gershenwald, Sapna P. Patel, Adi Diab, Isabella C. Glitza, Hussein Tawbi, Alexander J. Lazar, Patrick Hwu, Wen-Jen Hwu, Scott E. Woodman, Rodabe N. Amaria, Victor G. Prieto, and Jennifer A. Wargo enrolled subjects and contributed samples. *phs001493.v1.p1.c1* This study was supported by an AACR KureIt grant. *phs000980.v1.p1.c1.* We thank the members of the Thoracic Oncology Service and the Chan and Wolchok labs at MSKCC for helpful discussions, as well as the Immune Monitoring Core at MSKCC, including L. Caro, R. Ramsawak, and Z. Mu, for exceptional support with processing and banking peripheral blood lymphocytes. We thank P. Worrell and E. Brzostowski for help in identifying tumor specimens for analysis. We thank A. Viale for superb technical assistance. We thank D. Philips, M. van Buuren, and M. Toebes for help performing the combinatorial coding screens. This work was supported by the Geoffrey Beene Cancer Research Center (MDH, NAR, TAC, JDW, AS), the Society for Memorial Sloan Kettering Cancer Center (MDH), Lung Cancer Research Foundation (WL), Frederick Adler Chair Fund (TAC), The One Ball Matt Memorial Golf Tournament (EBG), Queen Wilhelmina Cancer Research Award (TNS), The STARR Foundation (TAC, JDW), the Ludwig Trust (JDW), and a Stand Up To Cancer-Cancer Research Institute Cancer Immunology Translational Cancer Research Grant (JDW, TNS, TAC). Stand Up To Cancer is a program of the Entertainment Industry Foundation administered by the American Association for Cancer Research. *phs001469.v1.p1.c1.* This work was supported by NIH grants R35CA197633, P01CA168585, 5P50CA168536, and GM08042. A comprehensive description of the dataset can be found at PMID:29320474. *phs001519.v1.p1.c1.* We thank the Ben and Catherine Ivy Foundation, the Blavatnik Family Foundation, the Broad Institute SPARC program, and NIH (NCI-1RO1CA155010-02 (to C.J.W.)), NHLBI-5R01HL103532-03 (to C.J.W.), Francis and Adele Kittredge Family Immuno-Oncology and Melanoma Research Fund (to P.A.O.), Faircloth Family Research Fund (to P.A.O.), NIH/NCI R21 CA216772-01A1 (to D.B.K.), NCI-SPORE-2P50CA101942-11A1 (to D.B.K.); NHLBI-T32HL007627 (to J.B.I.); NCI (R50CA211482) (to S.A.S.), Zuckerman STEM Leadership Program (to I.T.); Benoziyo Endowment Fund for the Advancement of Science (to I.T.); P50 CA165962 (SPORE) and P01 CA163205 (to K.L.L.); DFCI Center for Cancer Immunotherapy Research fellowship (to Z.H.); Howard Hughes Medical Institute Medical Research Fellows Program (to A.J.A.); and American Cancer Society PF-17-042-01–LIB (to N.D.M.). C.J.W. is a scholar of the Leukemia and Lymphoma Society. We thank the Center for Neuro-Oncology, J. Russell and Dana-Farber Cancer Institute (DFCI) Center for Immuno-Oncology (CIO) staff; B. Meyers, C. Harvey and S. Bartel (Clinical Pharmacy); M. Severgnini, K. Kleinsteuber, and E. McWilliams, (CIO laboratory); M. Copersino (Regulatory Affairs); T. Bowman (DFHCC Specialized Histopathology Core Laboratory); A. Lako (CIO); M. Seaman and D. H. Barouch (BIDMC); the Broad Institute's Biological Samples, Genetic Analysis and Genome Sequencing Platforms; J. Petricciani and M. Krane for regulatory advice; B. McDonough (CSBio), I. Javeri and K. Nellaiappan (CuriRx) for peptide development. *phs001565.v1.p1.c1* The research reported in this article was supported by BroadIgnite, BroadNext10, NIH K08CA188615, the Howard Hughes Medical Institute, and Stand Up To Cancer—American Cancer Society Lung Cancer Dream Team Translational Research Grant (grant number: SU2C-AACR-DT17-15). Stand Up To Cancer is a program of the Entertainment Industry Foundation. Research grants are administered by the American Association for Cancer Research, the scientific partner of SU2C.

## Author contributions

Original concept, R.M.P.; project supervision, H.C. and M.Z.; project planning and experimental design, A.C., R.M.P., C.P.D., M.Z., and H.C.; statistical advising, X.Z., W.K.T.; data acquisition, processing, and analysis, A.C. and R.M.P.; mutational signature analysis, L.A.; preparation of paper, A.C., R.M.P., M.Z., and H.C.

## Competing interests

R.M.P. is an employee and holds stock in Personalis. The remaining authors declare no competing interests.
