## [Peer Review File · Nature Communications]

Reviewers' comments:

Reviewer #1 (Remarks to the Author): expertise in sex differences in cancer and bioinformatics

In their manuscript, "Strength of Immune Selection in Tumors Varies with Sex and Age," Castro et al., investigate the mechanistic basis for the significant sex- and age- differences in response to checkpoint blockade amongst patients with melanoma. The published experience indicates that younger and female patients exhibit the poorest response to checkpoint blockade. The basis for this important finding has not been elucidated. Using their novel and validated Patient Harmonic-mean Best Rank score for evaluating the strength of patient MHC I- and II- specific somatic mutation presentation, the authors demonstrate, for the first time, that driver mutations present in younger and female melanoma patients are predicted to be weakly presented neo-antigens. This is hypothesized to be the basis for poor response to checkpoint blockade. This hypothesis, together with its corollary, that robust immuno-editing of clones with more antigenic driver mutations occurs more frequently in younger and female patients, is fascinating, important and timely. Overall, the study is very well done and clearly presented. The manuscript is well-written. I have only a few suggestions and comments.

1. Immuno-editing as a tumor control mechanism is not likely to be confined to neo-epitopes in driver mutations and among the great potential therapeutic values of checkpoint blockade is the ability to treat cancers with poorly targetable driver mutations. This may be particularly important in melanoma, which of course has the highest mutational burden of analyzed cancers and for which there are data to indicate that response to checkpoint blockade correlates with total mutational burden. I recognize that a focus on driver mutations is both important and practical as an approach to evaluating the importance of immuno-editing in cancer progression and response to checkpoint blockade, but I think the discussion would be strengthened by consideration of how this mechanism might generalize to immune response to non-driver mutations, how this might impact on the importance of different mutational signatures and exposures. It would be interesting to determine whether there are differences in predicted antigenicity of driver versus non-driver mutations.

2. Would the authors be willing to comment in the discussion on whether their analysis has relevance to pediatric cancers where youth and low mutational burdens may be negative predictors of response to checkpoint inhibition. Checkpoint inhibition has been in clinical trial for pediatric cancers for several years and as yet, there have been no exciting results reported like those in melanoma and lung cancer.

3. Figure 3panel A: Asterisks are associated with seven signatures (01,02,05,07a,07b,15,17a). Figure 3B indicates that 6 signatures are present in the evaluated driver mutations (01,02,05,10a,10b,13). Only three of these overlap (01,02,05) The text says 4 signatures overlap.

4. Line 105: The total numbers of patients excluded because of incompatibility with NetMHCpan affinity prediction should be indicated.

5. Line 106: "diverse in sex" really doesn't make sense. It would be more informative to indicate whether the ratio of male to female cases is consistent with the cancer population statistics from SEER. This would support the representativeness of the cohort.

Josh Rubin

Washington University School of Medicine

Reviewer #2 (Remarks to the Author): expertise in immunoediting

This paper reports an interesting finding: that immune selective pressure on driver mutations in tumors is stronger in younger patients and female patients, such that tumors from younger and female patients are "more invisible" to the immune system at the time treatment is initiated. To

reach this conclusion, the authors have applied their previously-published PHBR-I and -II scoring algorithms to the driver mutations present in tumors from TCGA patients. The conclusion this paper reaches is of particular interest as it offers a potential mechanism for the poorer outcomes to immunotherapy observed in younger and female patients that, if true, would be worthy of follow-up in additional studies.

My main critique of this paper is that, while the claims are intriguing, at present all the findings rely solely on differences in PHBR scores between patient groups. The PHBR scoring algorithm is a powerful tool, but it is still a single algorithm, and a relatively new one. For this paper to be suitable for publication in Nature Communications, I would expect to see additional data/findings corroborating the prediction of stronger immune selection in these patient groups, beyond just higher PHBR scores, to demonstrate robustness of this finding.

To address this concern, for example, the authors might consider looking at some of the drivers that they previously reported to be nearly-universally well-presented or nearly-universally poorly-presented by all MHC alleles. Are the most-well-presented driver mutations underrepresented in female patients and/or younger patients (in relevant indications)? And in contrast, are the most-poorly-presented driver mutations either equally abundant, or selectively enriched in males/older patients?

Additionally, several of the figures are difficult to read or interpret at present. Additional comments, including comments about these figures:

1. The data in Fig. S2 shows the average number of driver mutations/ patient to be < 1 . This would indicate that some patients have 0 driver mutations (at least, 0 from the list of 1,018). How does this impact PHBR scores? Are these patients simply excluded since they have no scorable mutations? (And if so, how many patients are not being scored?) Also, does the number of patients with "no drivers" differ between female/male and young/old cohorts?
2. It would also be helpful for the authors to comment on how a patient with multiple driver mutations is scored – in Fig 2 & 4 and related calculations, if a patient has 2 or 3 driver mutations, are those mutations counted as independent data points? Or is the most-visible one used? Or an average? Given that the claims of this paper at present rest very heavily on the findings reported in these two figures, it would seem these decisions, along with the decisions about how to handle patients with zero drivers, are extremely important and should be explained. Also, it would be useful for the authors to show how the distribution of number of driver mutations compares between cohorts, in addition to the data presented that the averages are not different in Fig. S2 and S7.
3. In supplementary Fig. S2, the authors report the average # driver mutations per patient group. It appears that female patients have a lower average # of drivers vs males, and similarly younger patients have a lower # vs. older. There isn't a p-value, nor data point dots (which would be helpful here, to see the range of drivers/patients) – the authors say this difference is not significant in the text, but would be useful to have the above information on this graph, and have the p-value in the caption or text.
4. The boxplot figures throughout the paper are hard to read, most noticeably Figs. 1A, 1C, 2A, and 3C. It is difficult to see which cohort has a higher mean (especially in 3C – the green and yellow means in MHC-II column look identical, despite an incredibly large reported p-value). Perhaps removing the display of the outlier data points (if this is acceptable to the journal) would enable better visibility of the essential part of the figures. Or alternatively, data could be plotted as violin plots as in Fig. S4. It would also be helpful to add the numeric values of these averages in the text where the figures are presented (or in the figure/legend). This would also enable better clarity on how different the PHBR scores are between females/males etc.

5. Fig. 3 is hard to follow. In Fig. 3A, which are the "4 signatures" that show sex-specific differences? And what do the asterisks mean? There are 8 asterisks in 7 columns, so it seems these do not identify the "4 signatures" mentioned in the text, but it is unclear what they do signify.

6. Fig. 3B is also difficult to interpret. The text referring to this figure says "only 4 of the signatures where sex-specific differences were observed contribute to the set of driver mutations used for this analysis". First, in this figure, there are 5 columns... are these 5 (?) signatures the ones that showed sex-specific differences? (They do not obviously appear to be so in panel 3A, particularly 10b). Second, the data here shows ~45% of all driver mutations belonging to signature 5, and another ~20% belonging to signature 1, both of which appear to have some of the greatest sex-specific variance in 3A. I find it hard to follow how this supports their claim that there is a low impact of sex-specific differences. (It also seems strange to me that these same 2 signatures are the ones associated with aging? Are the aging-associated signatures the same ones that show sex-specific differences?) The authors need to clarify what is being presented in this figure, and how it is to be interpreted, as it is at present confusing to follow.

7. In Fig. S6, showing PBHR score enrichment by cancer type, the shading of the boxes is difficult to interpret. There are a handful of dark-colored boxes where the result is easy to see. However the majority of these boxes are lightly shaded – which of these represent a significant result? (The text says "only melanoma" for S6A – is that uveal melanoma (the darkest box) or the "melanoma" column, which is much less obviously dark red?) Were any of the age-related results significant? Also the scale is unlabeled – the figure caption says "ratio" but there are negative values on the scale. Also, what is a "neutral" result? It seems tempting to interpret the grey-color boxes (e.g., bladder cancer boxes in S6A) as a neutral result/no enrichment - these seem correspond to a score of "0.8" on the scale, is that value actually indicative of no enrichment?

8. Fig 5 – The inclusion of a schematic/model is useful, but it does not appear that this figure accurately captures the reported findings. In this figure, at the time of diagnosis/treatment, it appears older male patients have twice as many driver mutations on their tumors as young female patients. The authors' data suggests that this is not at all what happens – rather, young female patients have a similar number of drivers, but different less-visible drivers, as older males. This figure should reflect the actual finding.

Reviewer #3 (Remarks to the Author): expertise in immunogenomics

Review of Castro et al

This is a well executed and timely piece of work which was well written and I found very interesting. My reservations are mainly linked to extrapolations in ICB sensitivity with data from tumours that are not ICB sensitive (as per most of the TCGA datasets). However, importantly their data does show sex specific pressures in NSCLC and melanoma. I have a number of comments which might improve the manuscript in its final form.

Points (major highlighted):

i) The authors refer to data suggesting in melanoma that females and younger individuals are less likely to respond to ICB - it should be noted that these are meta-analyses across multiple tumours. From both perspective of melanoma & NSCLC, the tumours most treated with ICB, there is divergence in presentation of tumours in both anatomical location (melanomas on limbs/ distal vs. back/ head males; NSCLC - tendency to be more proximal bronchi tumours, reduced smoking history). This doesn't negate the authors point – and could be seen to add to it – but, even given the high variance in cancer presentation, these tumours show very marked sex specific differences from the outset. Likewise, melanoma in younger individuals has a tendency to have lower BRAF

mutation rates and have more a heritable component.

ii) It would be preferable to express P values to 2 s.f. – e.g. 1a) $P=2.7e-4$ as opposed to 0.000265 across all figures. Similarly, if you are going to quote semi-exact P values for one figure (e.g. Fig. 1) then it would be preferable to be consistent in the approach across the paper (at least the main figures).

iii) Figure S2 – I am told that there are similar numbers of driver mutations across - but then shown barplots without indication as to range or evidence of statistical appraisal. Unless a good reason exists as to why not possible, it would be nice to see Figure S2 as boxplot with stats (as per all others).

iv) Figure S6 scale is confusing – if you are to use a colour scale then an effort should be made to centre this – otherwise false impressions are created. Thus, for the sex specific effect it should be centred on 0 – and this should be white (or at least not blue or red). As it stands, bladder is light grey/ pink – is there a sex effect here? If there was not I would expect it to be blue according to scale. Similarly for the age effect please centre on a neutral colour and keep the reds and greens for opposite ends of spectrum.

v) Figure 3c – worth altering quoted $P <$ in MHC-I panel – they are not <0

vi) Major: With respect to the assessment across tumour subtypes (mutational burden and sex assessment (figure 3)), the authors are rather underpowered to say much for many tumours – e.g. they use 17 tumours (10M, 7 F) for uveal melanoma. The observations from these small samples tend to be extreme but not significant – reflecting this lack of power. I think they would benefit from being more circumspect here and conserving this very worthwhile analysis to the tumours they have power in - the differences in mutational profiles in males versus females for melanoma for example being potentially consistent with the divergent natural history. By including so many negative observations (for which they are underpowered anyhow) this interesting fine detail is obscured.

vii) The natural history of melanoma, prior to treatment with ICB was that risk factors for recurrence and also death in the metastatic setting are sex (male worse than female) and age - although this effect is not so consistent. Therefore, these are the groups that have most to gain from ICB.

viii) Page 16 line 280:

“ Taken together, all evidence suggests that younger females are prone to stronger immunoediting than older males”.

No. Taken together, the evidence suggests that tumours developing within (younger) females are prone to stronger immunoediting than in (older) males. The data suggests the age effects are independent so conflating these two aspects is perhaps confusing.

Further questions:

i) Major: If the authors are correct in that in younger individuals and females the immune selection is greater, then just as qualitatively more immunogenic mutants more likely to be presented are less frequently found in these individuals – there should similarly be divergent quantitative expression of these mutated genes on an age/ sex basis. Do they observe this? It would be very nice to explore this aspect.

Reviewer #1 (Josh Rubin, Washington University School of Medicine) (Remarks to the Author):
expertise in sex differences in cancer and bioinformatics

In their manuscript, “Strength of Immune Selection in Tumors Varies with Sex and Age,” Castro et al., investigate the mechanistic basis for the significant sex- and age- differences in response to checkpoint blockade amongst patients with melanoma. The published experience indicates that younger and female patients exhibit the poorest response to checkpoint blockade. The basis for this important finding has not been elucidated. Using their novel and validated Patient Harmonic-mean Best Rank score for evaluating the strength of patient MHC I- and II- specific somatic mutation presentation, the authors demonstrate, for the first time, that driver mutations present in younger and female melanoma patients are predicted to be weakly presented neo-antigens. This is hypothesized to be the basis for poor response to checkpoint blockade. This hypothesis, together with its corollary, that robust immuno-editing of clones with more antigenic driver mutations occurs more frequently in younger and female patients, is fascinating, important and timely. Overall, the study is very well done and clearly presented. The manuscript is well-written. I have only a few suggestions and comments.

1. Immuno-editing as a tumor control mechanism is not likely to be confined to neo-epitopes in driver mutations and among the great potential therapeutic values of checkpoint blockade is the ability to treat cancers with poorly targetable driver mutations. This may be particularly important in melanoma, which of course has the highest mutational burden of analyzed cancers and for which there are data to indicate that response to checkpoint blockade correlates with total mutational burden. I recognize that a focus on driver mutations is both important and practical as an approach to evaluating the importance of immuno-editing in cancer progression and response to checkpoint blockade, but I think the discussion would be strengthened by consideration of how this mechanism might generalize to immune response to non-driver mutations, how this might impact on the importance of different mutational signatures and exposures. It would be interesting to determine whether there are differences in predicted antigenicity of driver versus non-driver mutations.

We agree with the reviewer. The immune system is not likely to discriminate between mutations and any presented mutation should be able to generate an anti-tumor immune response provided it is recognized by a T cell. One question that has been raised however, is whether the prevalence of a mutation across the tumor cells (e.g. clonal versus subclonal mutation) or dependence of the tumor cells on the mutation for survival (e.g. oncogene addiction) could influence whether the tumor can subsequently evade immune elimination. McGranahan *et al.* suggested that a clonal driver neoantigen is necessary for effective tumor elimination in response to immune checkpoint blockade [1]. The relevance to mutational signatures/exposures may then depend on when these signatures contribute mutations during tumor development, and whether the mutations they generate are biased in terms of potential to overlap with peptides presented by the MHC. We have added this conjecture to the discussion as follows:

- *“In this analysis, we focused on a set of recurrent missense and indel mutations in established driver genes developed in our previous work. This is motivated by the assumption that these are more likely to occur early during tumorigenesis, and may thus provide a view of immune selection before various mechanisms of immune evasion occur [2,3]. However it is unlikely that immune selection operates differently on different categories of mutation, and non-driver mutation derived neoantigens should be equally capable of triggering a T cell response. Whether tumor cells can evade T cell responses more easily when they are targeted against non-essential non-driver mutations remains an important question. It has been suggested that ICB responses are most effective when a clonal driver neoantigen is present [1]. While we did not observe large sex or age biases in the mutational signatures associated with the 1018 driver mutations, we speculate that it is possible non-driver mutations could show differences in their potential to serve as neoantigens if the underlying mutational processes are active at different times or are biased to generate mutations in expressed protein coding sequences with characteristics that bias their presentation.”*

As suggested by the reviewer, we further evaluated whether there are differences in predicted antigenicity of driver versus non-driver mutations. Because there are no adequate patient-specific antigenicity prediction tools, *i.e.* tools that can predict the activation of CD8 or CD4 T cells by a given peptide-MHC complex, we used peptide-MHC affinity as a proxy. We assume that the longer a peptide is able to remain bound on the cell surface, the higher the likelihood that a T cell will recognize it, and higher affinity should translate to longer cell surface exposure. Our previous work suggested that driver mutations occurring early during tumorigenesis were subject to stronger immune constraints than the same mutations occurring late (Figure R1 reproduced from Pyke et al 2018 [2]). We therefore compared the presentation of early and late occurring mutations divided according to their status as driver or passenger events. We used mutations observed in only a single tumor and in genes not implicated as cancer genes to represent passenger mutations. We designated mutations as occurring early or late using DNA VAF as described previously (Pyke et al 2018).

We found, as in Pyke et al., 2018, that earlier driver mutations tend to be significantly more poorly presented (have overall higher PHBR scores for) by patients' respective MHC alleles compared to late drivers as well as non-driver mutations. We observe the same trend for non-drivers, in that early occurring non-driver mutations had overall poorer affinity for a patient's MHC alleles. This analysis also suggests that non-driver mutations in each category were better presented than the corresponding driver mutations. This could support the hypothesis that tumor cells are less able to evade immune surveillance against driver mutations because the tumor cells are more dependent on driver mutations (Figure R2 A-B). However, studies have indicated that immune selection on non-drivers is more evident when considering only expressed mutations [4]. When only expressed passenger mutations were considered (Figure R2 C-D), we no longer saw a significant difference between early and late passengers while in general, all drivers remained more poorly presented than passengers for MHC Class I and early

drivers were more poorly presented than any passengers for Class II. This would seem to support that tumors are less able to avoid immune destruction targeted toward mutations that are under positive selection.

Figure R1. From Pyke et al., 2018: The MHC-I and MHC-II odds ratios (circles) and 95% confidence intervals (CIs) associated with a 1-unit increase in log PHBR-II score. Results are shown for mutations with low allelic fraction (dark gray) and high allelic fraction (light gray). Bars show 95% CIs.

Figure R2. Violinplots comparing the distribution of PHBR scores for early driver (blue) and late driver (orange) versus early non-driver (green) and late non-driver (red) mutations for Class I and Class II for (A-B) observed mutations and (C-D) expressed observed mutations. * indicates p-values <0.05 and ** indicates p-values <0.001.

1. McGranahan N, Furness AJS, Rosenthal R, Ramskov S, Lyngaa R, Saini SK, et al. Clonal neoantigens elicit T cell immunoreactivity and sensitivity to immune checkpoint blockade. *Science*. 2016;351: 1463–1469.
2. Marty Pyke R, Thompson WK, Salem RM, Font-Burgada J, Zanetti M, Carter H. Evolutionary Pressure against MHC Class II Binding Cancer Mutations. *Cell*. 2018;175: 1991.
3. Marty R, Kaabinejadian S, Rossell D, Slifker MJ, van de Haar J, Engin HB, et al. MHC-I Genotype Restricts the Oncogenic Mutational Landscape. *Cell*. 2017;171: 1272–1283.e15.
4. Yang F, Kim D-K, Nakagawa H, Hayashi S, Imoto S, Stein L, et al. Quantifying immune-based counterselection of somatic mutations. *PLoS Genet*. 2019;15: e1008227.

2. Would the authors be willing to comment in the discussion on whether their analysis has relevance to pediatric cancers where youth and low mutational burdens may be negative predictors of response to checkpoint inhibition. Checkpoint inhibition has been in clinical trial for

pediatric cancers for several years and as yet, there have been no exciting results reported like those in melanoma and lung cancer.

We believe that the community should be cautious about drawing comparisons between adult and pediatric cancers. Pediatric tumors tend to have very low mutation burdens [5–7], more germline factors and a more prevalent role for fusion genes and recurrent structural variants, likely leading to far fewer neoantigens overall. In addition, it is possible that somatic mutations occurring early enough during development could be perceived by the immune system as germline mutations.

We have added the following statement to the discussion: “Although our analysis suggests that younger age is associated with stronger anti-tumor immune responses, we strongly suggest caution in considering whether this trend could generalize to pediatric tumors. The genomic landscape of pediatric tumors is distinct from that of adulthood tumors, with lower mutation burdens, different driver events and more germline factors. Furthermore, the characteristics of the pediatric immune system differ greatly from those of an adult [8]. “

5. Vogelstein B, Papadopoulos N, Velculescu VE, Zhou S, Diaz LA Jr, Kinzler KW. Cancer genome landscapes. *Science*. 2013;339: 1546–1558.
6. Gröbner SN, Worst BC, Weischenfeldt J, Buchhalter I, Kleinheinz K, Rudneva VA, et al. The landscape of genomic alterations across childhood cancers. *Nature*. 2018;555: 321–327.
7. Ma X, Liu Y, Liu Y, Alexandrov LB, Edmonson MN, Gawad C, et al. Pan-cancer genome and transcriptome analyses of 1,699 paediatric leukaemias and solid tumours. *Nature*. 2018;555: 371–376.
8. Simon AK, Hollander GA, McMichael A. Evolution of the immune system in humans from infancy to old age. *Proc Biol Sci*. 2015;282: 20143085.

3. Figure 3 panel A: Asterisks are associated with seven signatures (01,02,05,07a,07b,15,17a). Figure 3B indicates that 6 signatures are present in the evaluated driver mutations (01,02,05,10a,10b,13). Only three of these overlap (01,02,05) The text says 4 signatures overlap.

We thank the reviewer for catching this mistake. We have corrected it to “3 signatures (01, 02, 05)”.

4. Line 105: The total numbers of patients excluded because of incompatibility with NetMHCpan affinity prediction should be indicated.

We have updated the text to include the numbers of patients per Class I and Class II analysis that have been excluded: “After excluding 515 patients from Class I and 1,064 patients from Class II analyses due to patients with HLA genotype incompatibility with NetMHCpan affinity

prediction software, 9,913 patients with MHC-I calls and 7,174 patients with MHC-II calls remained.”

5. Line 106: ‘diverse in sex’ really doesn’t make sense. It would be more informative to indicate whether the ratio of male to female cases is consistent with the cancer population statistics from SEER. This would support the representativeness of the cohort.

We thank the reviewer for this suggestion. Because it was not clear what adjustment or normalization went into the SEER “All Cancers” population statistics, we manually selected all tumor types from SEER (<https://seer.cancer.gov/statfacts/>) that were present in our TCGA cohort. As in the manuscript, we excluded sex-specific cancers before assessing the counts of male and female patients. We find that there is no statistically significant difference (Fisher’s exact, OR: 0.96, $p < 0.76$) between the SEER male/female ratio and that of our cohort (Figure R3). We have updated Figure S1 accordingly.

Figure R3. Barplot showing the female/male counts from SEER and the TCGA cohort used in this study, excluding sex-specific tumor types.

Reviewer #2 (Remarks to the Author): expertise in immunoediting

This paper reports an interesting finding: that immune selective pressure on driver mutations in tumors is stronger in younger patients and female patients, such that tumors from younger and female patients are “more invisible” to the immune system at the time treatment is initiated. To reach this conclusion, the authors have applied their previously-published PHBR-I and -II scoring algorithms to the driver mutations present in tumors from TCGA patients. The conclusion this paper reaches is of particular interest as it offers a potential mechanism for the poorer outcomes to immunotherapy observed in younger and female patients that, if true, would be worthy of follow-up in additional studies.

My main critique of this paper is that, while the claims are intriguing, at present all the findings rely solely on differences in PHBR scores between patient groups. The PHBR scoring algorithm is a powerful tool, but it is still a single algorithm, and a relatively new one. For this paper to be suitable for publication in Nature Communications, I would expect to see additional data/findings corroborating the prediction of stronger immune selection in these patient groups, beyond just higher PHBR scores, to demonstrate robustness of this finding.

To address this concern, for example, the authors might consider looking at some of the drivers that they previously reported to be nearly-universally well-presented or nearly-universally poorly-presented by all MHC alleles. Are the most-well-presented driver mutations underrepresented in female patients and/or younger patients (in relevant indications)? And in contrast, are the most-poorly-presented driver mutations either equally abundant, or selectively enriched in males/older patients?

We thank the reviewer for this excellent suggestion. To corroborate our finding of stronger immune selection in younger and female patients, we have analyzed the distribution of NetMHCpan affinity scores across observed driver mutations in our cohort. To be clear, the analysis presented below differs from our PHBR analysis in the manuscript in that we do not aggregate the affinity scores on a patient-specific basis, but simply use the median affinity score for the driver across all alleles available in NetMHCpan.

We find that more universally poorly presented driver mutations, both by MHC-I and MHC-II, are observed in younger and female patients than older and male patients (Figure R4-5).

In addition, in an analysis suggested by reviewer 3, we find that there is divergent quantitative RNA expression of these mutations on a sex and age basis (Figure R6), where females and younger patients have decreased fraction of RNA reads supporting the mutant allele. This provides additional evidence independent of the PHBR scoring framework that tumors in female and younger patients are on average subject to stronger immune selection than male and older patients.

Figure R4. Boxplots showing the distribution of median NetMHCpan Class I affinity scores for observed, expressed driver mutations across sex and age in our cohort. * indicates p-values <0.05 and ** indicates p-values <0.001.

Figure R5. Boxplots showing the distribution of median NetMHCpan Class II affinity scores for observed, expressed driver mutations across sex and age in our cohort. * indicates p-values <0.05 and ** indicates p-values <0.001.

Figure R6 (Copied from reviewer 3) Sex- and age-specific analysis of mutation RNA fraction. Box plots showing the distribution of fraction of RNA reads supporting the mutated allele in (A) female and male

patients, (B) younger and older patients, and (C) integrated sex- and age-specific patient cohorts. * indicates p-values <0.05 and ** indicates p-values <0.001.

Additionally, several of the figures are difficult to read or interpret at present. Additional comments, including comments about these figures:

1. The data in Fig. S2 shows the average number of driver mutations/ patient to be < 1. This would indicate that some patients have 0 driver mutations (at least, 0 from the list of 1,018). How does this impact PHBR scores? Are these patients simply excluded since they have no scorable mutations? (And if so, how many patients are not being scored?) Also, does the number of patients with “no drivers” differ between female/male and young/old cohorts?

We thank the reviewer for pointing out this potential source of confusion. Indeed, if patients did not have a driver mutation, they were excluded from the analyses in Figures 1-2 and Table 1. Figures 1 and 2 describe PHBR score distributions for observed, expressed driver mutations (Figures 1-2). Table 1 describes the results of linear models where we analyzed the effects of sex and age on the relationship between the probability of observing a driver mutation conditional on the potential to present that driver mutation.

We have sought to clarify this in the manuscript as follows:

- *Main text: For sex- and age-specific groups in each cohort, we compared the PHBR-I and PHBR-II score distributions for observed, RNA-expressed driver mutations observed in patient tumors, excluding 4,782 patients with no drivers from the list of 1018.*
- *Main text: We constructed sex- and age-specific generalized additive models with random effects to account for variation in mutation rate across individuals, and examined the coefficients corresponding to independent and interaction effects for PHBR-I, PHBR-II, and sex or age to assess their contribution to immune selection for expressed mutations observed ≥ 2 times in the cohort, excluding patients with no observed, expressed driver mutations.*

We did not observe a bias in the number of observed driver mutations with respect to age or sex alone (See response to next comment, Figure R9). However, when we compared the number of patients with no driver mutations from the list of 1018 across sex and age by comparing samples in the lower 30th and higher 70th percentile of the age distribution, (Table R7) we found that younger female patients were overrepresented in the group with no observed drivers. (MHC-I cohort: OR=1.12, p<0.12, MHC-II cohort: OR=1.28, p<0.015). Specifically, we analyzed the 4,782 confidently typed, microsatellite stable patients from non-sex-specific tumor types that did not have a driver from our analysis of expressed, observed mutations that were omitted from (Figures 1-2). We note that there was an overrepresentation of thyroid cancer cases among the younger females with no driver mutations (Figure R8). This is a disease associated with low mutational burden and that typically only has a single driver mutation [9]. This is consistent with other analyses, including the observation of divergent alternate allele expression found based

on suggestion from reviewer 3 (Figure R6, previous question), where younger and female patients also had lower expression of their driver mutations, presumably due to greater immune selection in female patients.

This finding is now included in the manuscript as follows:

- “While the number of observed drivers was not significantly different between sex and age groups (Figure S2), younger female patients were overrepresented in the group with no observed driver mutations (Fisher’s exact test: Class I: OR=1.12, $p<0.12$; Class II: OR=1.28, $p<0.015$). We note this group had an overrepresentation of thyroid cancer cases, a disease associated with low mutational burden and that typically only has a single driver mutation.”

9. Cancer Genome Atlas Research Network. Integrated genomic characterization of papillary thyroid carcinoma. Cell. 2014;159: 676–690.

	Class I		Class II	
	Younger	Older	Younger	Older
Female	279	288	436	366
Male	343	422	417	563
	FE: OR=1.12, $p<0.12$		FE: OR=1.28, $p<0.015$	

Table R7. Counts and Fisher’s exact results comparing the numbers of patients with no driver mutations.

Figure R8. Heatmaps showing the number of female/male and younger/older patients with no driver mutation by tumor type for (top) Class I and (bottom) Class II.

2. It would also be helpful for the authors to comment on how a patient with multiple driver mutations is scored – in Fig 2 & 4 and related calculations, if a patient has 2 or 3 driver mutations, are those mutations counted as independent data points? Or is the most-visible one used? Or an average? Given that the claims of this paper at present rest very heavily on the findings reported in these two figures, it would seem these decisions, along with the decisions about how to handle patients with zero drivers, are extremely important and should be explained. Also, it would be useful for the authors to show how the distribution of number of driver mutations compares between cohorts, in addition to the data presented that the averages are not different in Fig. S2 and S7.

In the boxplot figures (Figures 1,2,4), all driver mutations are included, therefore if a patient has multiple expressed driver mutations, all scores are included as independent data points. As the boxplot analysis was intended to demonstrate that the respective affinities of retained, expressed driver mutations were different across sex and age, we included all observations. To control for the fact that some patients had multiple driver mutations, and thus are not completely independent events, we included patient ID as a random effect in our linear model (Table 1), which ensured a more rigorous analysis of the interaction of sex and age with mutation affinity on the probability of mutation. We have clarified this in the methods and manuscript as follows:

- *Main text: (copied from above) For sex- and age-specific groups in each cohort, we compared the PHBR-I and PHBR-II score distributions for observed, RNA-expressed driver mutations observed in patient tumors, excluding 4,782 patients with no drivers from the list of 1018.*
- *Main text: We constructed sex- and age-specific generalized additive models with random effects to account for variation in mutation rate across individuals, and examined the coefficients corresponding to independent and interaction effects for PHBR-I, PHBR-II, and sex or age to assess their contribution to immune selection for expressed mutations observed ≥ 2 times in the cohort, excluding patients with no observed, expressed driver mutations. To control for the fact that some patients had multiple driver mutations, and thus are not completely independent events, we included patient ID as a random effect in our linear model.*
- *Methods: To estimate the effects of PHBR and sex or age on probability of mutation, we considered the following random effects models: ... where $\eta_i \sim N(0, \theta\eta)$ are random effects capturing different mutation propensities among patients, using patient IDs.*

To compare the discrete distributions of the number of driver mutations, we first computed the empirical cumulative distribution function (ECDF) (Figure R9) and then evaluated the difference

in distribution using the two-sample Kolmogorov-Smirnov statistic and found no statistical difference across sex or age. We have included this analysis in supplementary figure 2.

Figure R9. Empirical cumulative distributions of the total number of driver mutations per patient across (A-B) sex and (C-D) age (30/70th percentile thresholds for younger/older patients) in our cohort for Class I (left) and Class II (right).

3. In supplementary Fig. S2, the authors report the average # driver mutations per patient group. It appears that female patients have a lower average # of drivers vs males, and similarly younger patients have a lower # vs. older. There isn't a p-value, nor data point dots (which would be helpful here, to see the range of drivers/patients) – the authors say this difference is not significant in the text, but would be useful to have the above information on this graph, and have the p-value in the caption or text.

We thank the reviewer for this comment. We have updated the analysis to be more statistically rigorous and now use the ECDF distributions with p-values generated above (Figure R9) in supplementary figure S2.

4. The boxplot figures throughout the paper are hard to read, most noticeably Figs. 1A, 1C, 2A, and 3C. It is difficult to see which cohort has a higher mean (especially in 3C – the green and yellow means in MHC-II column look identical, despite an incredibly large reported p-value). Perhaps removing the display of the outlier data points (if this is acceptable to the journal) would enable better visibility of the essential part of the figures. Or alternatively, data could be plotted as violin plots as in Fig. S4. It would also be helpful to add the numeric values of these averages

in the text where the figures are presented (or in the figure/legend). This would also enable better clarity on how different the PHBR scores are between females/males etc.

We thank the reviewer for these suggestions. We have added the median values in figures 1, 2, and 4 and have zoomed in from y-axis range (0-100) to (0-60) for figure 3C. We have also replaced the “0” p-value for figure 3C with the minimum float value for Python following a suggestion from Reviewer 3.

5. Fig. 3 is hard to follow. In Fig. 3A, which are the “4 signatures” that show sex-specific differences? And what do the asterisks mean? There are 8 asterisks in 7 columns, so it seems these do not identify the “4 signatures” mentioned in the text, but it is unclear what they do signify.

We mistakenly put 4 signatures instead of 3 (signatures 01, 02, and 05), and have now rectified the error in the text, and added text clarifying which signatures: *“Importantly, only 3 of the signatures (01, 02, and 05) where we observed significant sex-specific differences contribute to the set of driver mutations used for this analysis (Figure 3B)”*

The asterisks indicate significant log ratios between males and females for mutational signatures in specific cancer types. We have updated the figure legend text to specify what the asterisks mean: *“Heatmap of log₂ male (blue) to female (pink) ratios of mutational signatures for each tumor type with asterisks denoting a significantly different ratio between genders”*.

6. Fig. 3B is also difficult to interpret. The text referring to this figure says “only 4 of the signatures where sex-specific differences were observed contribute to the set of driver mutations used for this analysis”. First, in this figure, there are 5 columns... are these 5 (?) signatures the ones that showed sex-specific differences? (They do not obviously appear to be so in panel 3A, particularly 10b). Second, the data here shows ~45% of all driver mutations belonging to signature 5, and another ~20% belonging to signature 1, both of which appear to have some of the greatest sex-specific variance in 3A. I find it hard to follow how this supports their claim that there is a low impact of sex-specific differences. (It also seems strange to me that these same 2 signatures are the ones associated with aging? Are the aging-associated signatures the same ones that show sex-specific differences?) The authors need to clarify what is being presented in this figure, and how it is to be interpreted, as it is at present confusing to follow.

We thank the reviewer for pointing out the need for clarification. As mentioned in the above response, we have corrected the text to say 3 signatures (01, 02, 05) instead of 4. Figure 3B shows the mutational signatures (05, 10b, 01, 02, 13, 10a) that could have generated the 1018 driver mutations based on flanking DNA context.

Signatures 01 and 05 are clocklike signatures associated with errors that occur during cell division, and are thus detected in all tumor types, whereas exposure-related and impaired DNA damage repair signatures are more often only observed in only a subset of tumors and in specific tumor types. This may contribute to the appearance that there is more sex associated variance in signatures 01 and 05. However, the sex difference in number of mutations attributable to these signatures is only statistically significant in two tumor types (LIHC: signatures 1 and 5; GBM: signature 1), and in these cases, the ratio is not very large (-0.51, 0.49, and -0.81, respectively).

We have clarified in the manuscript as following:

- *Importantly, only 3 of the signatures (01, 02, and 05) where we observed significant sex-specific differences contribute to the set of driver mutations used for this analysis (Figure 3B). Since signatures 01 and 05 are endogenous rather than exposure associated signatures, this suggests a very low impact of environmental exposures of sex-specific effects of immune selection on drivers. Furthermore, when we excluded the tumor types with significant signature differences (GBM and LIHC), we still observed sex- and age-related differences (Table S2).*

Updated Figure 3: Sex-specific exposure analysis with mutational signatures. (A) Heatmap of log2 male (blue) to female (pink) ratios of mutational signatures for each tumor type with asterisks denoting a significantly different ratio between male and female sexes. (B) The percentage of mutations in the set of driver mutations that are part of each mutational signature. (C) Boxplot comparing allele-specific MHC-I and MHC-II presentation scores of C>T or T>C driver mutations (green) versus driver mutations resulting from other base substitutions (yellow).

7. In Fig. S6, showing PBHR score enrichment by cancer type, the shading of the boxes is difficult to interpret. There are a handful of dark-colored boxes where the result is easy to see.

However the majority of these boxes are lightly shaded – which of these represent a significant result? (The text says “only melanoma” for S6A – is that uveal melanoma (the darkest box) or the “melanoma” column, which is much less obviously dark red?) Were any of the age-related results significant? Also the scale is unlabeled – the figure caption says “ratio” but there are negative values on the scale. Also, what is a “neutral” result? It seems tempting to interpret the grey-color boxes (e.g., bladder cancer boxes in S6A) as a neutral result/no enrichment - these seem correspond to a score of “0.8” on the scale, is that value actually indicative of no enrichment?

We thank the reviewer for this helpful comment. As per this reviewer’s and reviewer 3’s suggestion we have corrected the scale on this figure by log transforming the values and centering the scale at 0, and have excluded tumor types with fewer than 30 samples in each category to focus on the cases where we would be reasonably powered to detect a significant difference should it exist. As a result, uveal melanoma which has a very small sample size is no longer shown. The only significant results are in skin cutaneous melanoma, significance is now indicated with asterisks (Figure R10).

The grey “neutral” results indicate that the median affinity scores are the same or nearly the same across sex or age for that tumor type. We find as sample sizes get larger, the difference between medians gets smaller and the log2 ratio tends to get closer to 0 (*i.e.* more grey), suggesting that the true differences in affinity for each tumor type are likely to be subtle (Figure R11). Overall, this analysis reveals that the sex and age differences observed pan-cancer in our analysis are not driven by any one tumor type, and larger sample sizes will be required to detect significant differences in individual tumor types.

Figure R10. (Updated Figure S6) Heatmaps showing disease-specific PHBR-I and PHBR-II log2 median ratios for (A) females vs. males where red coloring indicates higher median female PHBR distributions, and (B) younger vs. older patients where green coloring indicates higher median younger PHBR distributions. “Age gap” shows disease-specific age thresholds (30th and 70th percentile), with darker coloring indicating a wider age gap and vice versa. A minimum of 30 patients in each group was required;

blanks indicate that fewer than 30 samples were available in one of the categories. ** indicates $p < 0.001$, * indicates $p < 0.05$, and • indicates $p < 0.1$.

Figure R11. Regression plots showing the trend for median PHBR ratio to decrease as mean sample size between respective groups in each heatmap category increases across (A-B) sex and (C-D) age.

8. Fig 5 – The inclusion of a schematic/model is useful, but it does not appear that this figure accurately captures the reported findings. In this figure, at the time of diagnosis/treatment, it appears older male patients have twice as many driver mutations on their tumors as young female patients. The authors' data suggests that this is not at all what happens – rather, young female patients have a similar number of drivers, but different less-visible drivers, as older males. This figure should reflect the actual finding.

We thank the reviewer for catching this unintended implication. We have updated the schematic to better reflect our reported findings:

Updated Figure 5: Proposed model of the relationship between immune selection and immunotherapy in cancer patients. Young females experience the strongest immune response, rendering their diagnosed tumors very invisible to the immune system and difficult to treat with ICB. On the other extreme, old males experience the weakest immune response, leaving their diagnosed tumors very visible to the immune system and open to attack when stimulated with ICB.

Reviewer #3 (Remarks to the Author): expertise in immunogenomics

This is a well executed and timely piece of work which was well written and I found very interesting. My reservations are mainly linked to extrapolations in ICB sensitivity with data from tumours that are not ICB sensitive (as per most of the TCGA datasets). However, importantly their data does show sex specific pressures in NSCLC and melanoma. I have a number of comments which might improve the manuscript in its final form.

Points (major highlighted):

i) The authors refer to data suggesting in melanoma that females and younger individuals are less likely to respond to ICB - it should be noted that these are meta-analyses across multiple tumours. From both perspective of melanoma & NSCLC, the tumours most treated with ICB, there is divergence in presentation of tumours in both anatomical location (melanomas on limbs/ distal vs. back/ head males; NSCLC - tendency to be more proximal bronchi tumours, reduced smoking history). This doesn't negate the authors point – and could be seen to add to it – but, even given the high variance in cancer presentation, these tumours show very marked sex specific differences from the outset. Likewise, melanoma in younger individuals has a tendency to have lower BRAF mutation rates and have more a heritable component.

We thank the reviewer for highlighting these important subtleties. We have revised the introduction and discussion accordingly:

- Intro: “Meta-analyses of clinical trials in multiple cancer types treated with ICB suggest that young and female patients are characterized by low response rates”
- Discussion: “This finding is consistent with recent meta-analyses across multiple tumors showing sex- and age-dependent differences in response to ICB.”

ii) It would be preferable to express P values to 2 s.f. – e.g. 1a) $P=2.7e-4$ as opposed to 0.000265 across all figures. Similarly, if you are going to quote semi-exact P values for one figure (e.g. Fig. 1) then it would be preferable to be consistent in the approach across the paper (at least the main figures).

We thank the reviewer for this suggestion and have updated Figure 1 in the manuscript (also shown below). We have also updated the p-values in Figure 3 and 4 to have two significant figures.

Updated Figure 1: Sex- and age-specific MHC presentation of observed, RNA-expressed driver mutations. (A-B) Box plots denoting the distribution of (A) PHBR-I and (B) PHBR-II scores for expressed driver mutations in female and male pan-cancer patients. (C-D) Box plots denoting the distribution of (C) PHBR-I and (D) PHBR-II scores for expressed driver mutations in younger and older pan-cancer patients. Median values are shown in white.

iii) Figure S2 – I am told that there are similar numbers of driver mutations across - but then shown barplots without indication as to range or evidence of statistical appraisal. Unless a good reason exists as to why not possible, it would be nice to see Figure S2 as boxplot with stats (as per all others).

In addressing a similar comment from Reviewer 2, we have compared the discrete distributions of the number of driver mutations, by first computing the empirical cumulative distribution function (ECDF) (Figure R9) and evaluating the difference in distribution using the two-sample Kolmogorov-Smirnov statistic. We found no statistical difference across sex or age and have replaced the barplots with this analysis in supplementary figure 2.

Figure R9. Empirical cumulative distributions of the total number of driver mutations per patient across (A-B) sex and (C-D) age in our cohort for Class I (left) and Class II (right).

iv) Figure S6 scale is confusing – if you are to use a colour scale then an effort should be made to centre this – otherwise false impressions are created. Thus, for the sex specific effect it should be centred on 0 – and this should be white (or at least not blue or red). As it stands, bladder is light grey/ pink – is there a sex effect here? If there was not I would expect it to be blue according to scale. Similarly for the age effect please centre on a neutral colour and keep the reds and greens for opposite ends of spectrum.

We thank the reviewer for catching this. We have updated the color scale (also in response to suggestion from reviewer 2) to be log-transformed and the color ranges now better reflect ratio values (*i.e.* centered around 0, which reflects no differences). As suggested below, we have also decided to focus on tumor types/analyses that are better powered (minimum 30 patients) to detect a difference if it exists (shown in response to comment vi). We only observe a significant difference for sex in melanoma where we have over 500 patients.

v) Figure 3c – worth altering quoted P< in MHC-I panel – they are not <0

We have updated the p-value to be “p<2e-308”, the minimum number displayable in Python.

vi) Major: With respect to the assessment across tumour subtypes (mutational burden and sex assessment (figure 3)), the authors are rather underpowered to say much for many tumours – e.g. they use 17 tumours (10M, 7 F) for uveal melanoma. The observations from these small samples tend to be extreme but not significant – reflecting this lack of power. I think they would

benefit from being more circumspect here and conserving this very worthwhile analysis to the tumours they have power in - the differences in mutational profiles in males versus females for melanoma for example being potentially consistent with the divergent natural history. By including so many negative observations (for which they are underpowered anyhow) this interesting fine detail is obscured.

This is a great point. We agree that we are underpowered to make any statements for many of the tumor types. We have updated the figure to only display results for analyses where we have a minimum of 30 patients in each category. As the reviewer suspected, focusing on larger sample size cases suggests effect sizes are likely to be small and much larger sample sizes may be needed to detect significance of any observed differences.

Updated Figure S6. Heatmaps showing disease-specific PHBR-I and PHBR-II median ratios for (A) females vs. males where red coloring indicates higher median female PHBR distributions, and (B) younger vs. older patients where green coloring indicates higher median younger PHBR distributions. “Age gap” shows disease-specific age thresholds (30th and 70th percentile), with darker coloring indicating a wider age gap and vice versa. A minimum of 30 patients in each group was required; blanks indicate that fewer than 30 samples were available in one of the categories. ** indicates $p < 0.001$, * indicates $p < 0.05$, and • indicates $p < 0.1$.

vii) The natural history of melanoma, prior to treatment with ICB was that risk factors for recurrence and also death in the metastatic setting are sex (male worse than female) and age - although this effect is not so consistent. Therefore, these are the groups that have most to gain from ICB.

We thank the reviewer for this comment, we have added the following to the conclusion:

“Notably prior to treatment with ICB, male sex (and less consistently older age) are associated with higher risk of recurrence and death in melanoma and may stand to benefit more from ICB,

thus it is also possible that overall stronger immune surveillance could prove advantageous in the context of ICB despite differences in the quality of neoantigens.”

viii) (Page 16 line 280:) ”Taken together, all evidence suggests that younger females are prone to stronger immunoediting than older males”. No. Taken together, the evidence suggests that tumours developing within (younger) females are prone to stronger immunoediting than in (older) males. The data suggests the age effects are independent so conflating these two aspects is perhaps confusing.

We thank the reviewer and agree with this suggestion. We have updated the text to clarify this point: “Taken together, the evidence suggests that tumors developing in younger and female patients are prone to stronger immunoediting than those in older and male patients.”

Further questions:

i) Major: If the authors are correct in that in younger individuals and females the immune selection is greater, then just as qualitatively more immunogenic mutants more likely to be presented are less frequently found in these individuals – there should similarly be divergent quantitative expression of these mutated genes on an age/ sex basis. Do they observe this? It would be very nice to explore this aspect.

We thank the reviewer for this excellent suggestion. We analyzed mutation-specific expression in the context of sex and age and found that female and younger patients’ mutations had lower fraction of RNA reads supporting the mutated base than males and older patients (Figure R12). This supports the hypothesis that in addition to differences in immunogenic mutations on a sex/age basis, there is similarly divergent expression of these mutations.

Figure R12. Sex- and age-specific analysis of mutation RNA fraction. Box plots showing the distribution of fraction of RNA reads supporting the mutated allele in (A) female and male patients, (B) younger and older patients, and (C) integrated sex- and age-specific patient cohorts. * indicates p-values <math>< 0.05</math> and ** indicates p-values <math>< 0.001</math>.

REVIEWERS' COMMENTS:

Reviewer #1 (Remarks to the Author):

I appreciate the thoughtfulness and thoroughness of the authors' responses to all the reviewer critiques. I have no further comments or concerns.

Reviewer #2 (Remarks to the Author):

This paper has been improved by the authors' revisions. In particular, the authors have addressed my main critique of relying solely on the PBHR algorithm to make their claims, and now include an analysis of NetMHCpan binding affinity scores of expressed drivers (Fig. S6) and of the fraction of RNA reads supporting mutant vs wildtype transcripts (Fig. S7). This greatly strengthens their findings. The authors' edits to the figures have also greatly improved their readability.

I have a few additional minor comments on the text, all of which I believe would be straightforward for the authors to address:

1. On page 8, the authors say that 4,782 patients are excluded due to not having one of the 1,018 drivers. It would be helpful if they would also state how many patients are actually being included in the driver analysis, since it seems a significant number (~50%?) are being excluded.
2. On page 9, the authors say "There was a general trend for female and younger tumors..." – this phrasing is awkward. It is the patients, rather than the tumors themselves, that are female/younger.
3. Figure S8 (formerly S6) has been greatly improved by the new color scale and the exclusion of indications with small numbers. In discussing the results of the age analysis (page 9), it seems an odd omission that the authors mention several cancer types that trend against their general finding of higher PBHR scores in younger patients, but do not mention rectal cancer which was the only one to reach significance ($p < 0.1$ by their legend) for higher PBHR scores in older patients.
4. The authors have added the medians in white to several figures, which improves their readability. It would be nice to have the medians also added to Fig. 3C and the boxplots in the supplementary figures (specifically S3, S6, & S7).
5. On page 13, the authors mention excluding GBM and LIHC from an analysis—these tumor types/acronyms should be spelled out (e.g., glioblastoma and liver hepatocellular carcinoma)
6. On page 15, the manuscript reads "simultaneous effects of sex and age (Figure 4E,D)"—this appears to be referencing Figure 4E, F, and not panel D.
7. The legend of Fig. S2 should include mention of panels C and D.

Reviewer #3 (Remarks to the Author):

The authors have greatly improved their manuscript in response to my comments and those of other reviewers. Specifically, they have fully addressed my comments and I feel the manuscript now warrants publication. I apologise for delays in returning this review.

REVIEWERS' COMMENTS:

Manuscript updates:

Notice of error correction

In preparing our revised manuscript, we identified and corrected an error in the statistical test calculations for Supplementary Figure 4 panels A-D and Supplementary Figure 9 panels E-H, describing patient potential to present driver mutations in the discovery and validation cohorts respectively. These panels compared the distribution of patient-specific presentation scores across all 1018 possible driver mutations. Previously, the wrong column was input to the Mann Whitney U statistical test when comparing all driver affinities between male/female and older/younger groups, and as a result, incorrect p-values were reported in the figure. The figures themselves have not changed, but we have now corrected the p-values which indicate statistical significance.

Original Supplementary Figure 4A-F

Updated Supplementary Figure 4A-F

Original Supplementary Figure 9E-J

Updated Supplementary Figure 9E-J

Although the test now shows significance, the effect sizes are very small (closer to 0) as compared with the main effects detected in the manuscript, (effect size quantified by Cliff's d which ranges from -1 to 1; Table R1). Furthermore, rather than resulting in p -values closer to 1, permuting the age/sex labels and repeating the test 1000 times actually results in p -values that get closer to 0. This suggests that the large number of data points is driving the significant p -values rather than true differences in the distributions (Figure R1). Finally, we observed no differences in the empirical cumulative distribution functions of scores representing fraction of presented drivers between groups (significance assessed by Kolmogorov-Smirnov test), already shown in Supplementary Figure 4 panels E-F and Supplementary Figure 9 panels I-J in the previous version of the manuscript. These ECDFs are also unchanged from the previous submission, though a legend and corresponding statistics have been added for clarity.

		PHBR-I (sex)	PHBR-II (sex)	PHBR-I (age)	PHBR-II (age)
Discovery	Figure 1 effect sizes (r)	-0.0654	-0.104	-0.081	-0.0734
	Supplementary Figure 4A-D effect sizes (r)	-0.00276	-0.00381	-0.00529	-0.00144

Validation	Figure 4 effect sizes (r)	-0.154	-0.164	-0.207	-0.346
	Supplementary Figure 9E-H effect sizes (r)	-0.016	-0.000938	-0.00132	-0.0212

Table R1. Effect sizes shown for figures 1A-D, 4A-D and supplementary figures 4A-D and 9E-H. Effect sizes are calculated using Cliff's d.

Considering these two factors (1) effect sizes are very small, even though the p-values are significant, and in any case not large enough to account for effect sizes detected in Figures 1 and 4 and (2) the KS test in Supplementary Figures 4E-F and 9I-J, which provides a more complete and relevant comparison as it is capable of detecting differences in presentation at any threshold, finds no difference, this change does not affect our interpretation of the results.

In addition to updating the figure, we have updated the main manuscript to reflect this change as follows:

- We then compared the distributions of patient PHBR-I and PHBR-II scores across the 1,018 driver mutations (Supplementary Figure 4A-D) and found significant p-values, but very small effect sizes between groups. To ensure that the potential to present driver mutations was consistent across sex and age, we compared the fraction of presented drivers at various score thresholds, and found no significant differences (Supplementary Figure 4E-F).
- Importantly, we found that these observed between-group differences in PHBR scores were far greater (falling outside the 99% confidence interval) than differences when we randomly reassigned mutations across patients and recalculated patient-specific PHBR scores (Methods; Supplementary Figure 5), **and were an order of magnitude greater than the effect sizes observed when comparing score distributions independent of mutation occurrence (Supplementary Figure S4).**

We have also added the effect sizes in Table R1 to figure legends. We apologize for not catching this error earlier.

Figure R1. Histograms showing p-value distributions for (top) sex and (bottom) age cohorts after comparing for differences in driver affinities for MHC I (left) and MHC II (right) between shuffled sex and age, respectively. P-values were calculated using the Mann Whitney U test. The vertical line shows the calculated p-value for the true labels.

Reviewer #1 (Remarks to the Author):

I appreciate the thoughtfulness and thoroughness of the authors' responses to all the reviewer critiques. I have no further comments or concerns.

Reviewer #2 (Remarks to the Author):

This paper has been improved by the authors' revisions. In particular, the authors have addressed my main critique of relying solely on the PBHR algorithm to make their claims, and now include an analysis of NetMHCpan binding affinity scores of expressed drivers (Fig. S6) and of the fraction of RNA reads supporting mutant vs wildtype transcripts (Fig. S7). This greatly strengthens their findings. The authors' edits to the figures have also greatly improved their readability.

I have a few additional minor comments on the text, all of which I believe would be straightforward for the authors to address:

1. On page 8, the authors say that 4,782 patients are excluded due to not having one of the 1,018 drivers. It would be helpful if they would also state how many patients are actually being included in the driver analysis, since it seems a significant number (~50%?) are being excluded.

We have now updated the text to state the number of patients included for the actual analysis.

Added text: We therefore performed sex-specific analysis for unique 2,900 patients and age-specific analysis for 3,928 unique patients.

2. On page 9, the authors say “There was a general trend for female and younger tumors...” – this phrasing is awkward. It is the patients, rather than the tumors themselves, that are female/younger.

We have updated the phrasing.

Updated text: There was a general trend for female and younger patients’ tumors to have higher median PHBR-I and II scores across tumor types, although the difference was only statistically significant in melanoma (Supplementary Figure 8A)

3. Figure S8 (formerly S6) has been greatly improved by the new color scale and the exclusion of indications with small numbers. In discussing the results of the age analysis (page 9), it seems an odd omission that the authors mention several cancer types that trend against their general finding of higher PBHR scores in younger patients, but do not mention rectal cancer which was the only one to reach significance ($p < 0.1$ by their legend) for higher PBHR scores in older patients.

We have updated the text to mention rectal cancer.

Updated text: ...with some notable exceptions that included rectal cancer, thyroid cancer, stomach cancer, and liver (Supplementary Figure 8B).

4. The authors have added the medians in white to several figures, which improves their readability. It would be nice to have the medians also added to Fig. 3C and the boxplots in the supplementary figures (specifically S3, S6, & S7).

We have added medians to Figure 3C, S3, S6, S7.

Updated figures:

3C

S3

S6

5. On page 13, the authors mention excluding GBM and LIHC from an analysis—these tumor types/acronyms should be spelled out (e.g., glioblastoma and liver hepatocellular carcinoma)
 We have now spelled these terms out.

Updated text: Furthermore, when we excluded the tumor types with significant signature differences (glioblastoma multiforme, GBM and liver hepatocellular carcinoma, LIHC), we still observed sex- and age-related differences (Table S2).

6. On page 15, the manuscript reads “simultaneous effects of sex and age (Figure 4E,D)”—this appears to be referencing Figure 4E, F, and not panel D.
We have updated the figure reference.

Updated text: When we examined the simultaneous effects of sex and age (Figure 4E,F),

7. The legend of Fig. S2 should include mention of panels C and D.
We have added mention of panels C and D.

Updated legend: The empirical cumulative distribution functions (ECDF) of driver mutations pan-cancer per cohort. Curves show the cumulative number of driver mutations in each sex- and age-specific cohort for microsatellite-stable patients with NetMHCpan-compatible (A,C) MHC-I and (B,D) MHC-II calls. P-values were obtained from the Kolmogorov-Smirnov two-sample test.

Reviewer #3 (Remarks to the Author):

The authors have greatly improved their manuscript in response to my comments and those of other reviewers. Specifically, they have fully addressed my comments and I feel the manuscript now warrants publication. I apologise for delays in returning this review.